# Exploring the Limits of Out-of-Distribution Detection

**Stanislav Fort**[*]
Stanford University
sfort1@stanford.edu

**Jie Ren**[*]
Google Research, Brain Team
jjren@google.com

**Balaji Lakshminarayanan**
Google Research, Brain Team
balajiln@google.com

## Abstract

Near out-of-distribution detection (OOD) is a major challenge for deep neural networks. We demonstrate that large-scale pre-trained transformers can significantly improve the state-of-the-art (SOTA) on a range of near OOD tasks across different data modalities. For instance, on CIFAR-100 vs CIFAR-10 OOD detection, we improve the AUROC from 85% (current SOTA) to 96% using Vision Transformers pre-trained on ImageNet-21k. On a challenging genomics OOD detection benchmark, we improve the AUROC from 66% to 77% using transformers and unsupervised pre-training. To further improve performance, we explore the few-shot outlier exposure setting where a few examples from outlier classes may be available; we show that pre-trained transformers are particularly well-suited for outlier exposure, and that the AUROC of OOD detection on CIFAR-100 vs CIFAR-10 can be improved to 98.7% with just 1 image per OOD class, and 99.46% with 10 images per OOD class. For multi-modal image-text pre-trained transformers such as CLIP, we explore a new way of using just the names of outlier classes as a sole source of information without any accompanying images, and show that this outperforms previous SOTA on standard vision OOD benchmark tasks.

## 1 Introduction

Deep neural networks are increasingly used in high-stakes applications such as healthcare [Roy et al., 2021, Ren et al., 2019]. Safe deployment of models requires that models not only be accurate but also be robust to distribution shift [Amodei et al., 2016]. Neural networks can assign high-confidence predictions to mis-classified inputs [Guo et al., 2017, Lakshminarayanan et al., 2017] as well as test inputs that do not belong to one of the training classes [Nguyen et al., 2015]. This motivates the need for methods that can reliably detect out-of-distribution (OOD) inputs. There has been a lot of progress in detecting OOD inputs including methods based on discriminative models [Hendrycks and Gimpel, 2016, Lee et al., 2018, Liang et al., 2017, Liu et al., 2020] as well as methods based on deep generative models [Nalisnick et al., 2019, Zhang et al., 2020].

The difficulty of the OOD detection task depends on how semantically close the outliers are to the inlier classes. Winkens et al. [2020] distinguish between *near-OOD* tasks which are harder and *far-OOD* tasks which are easier, as evidenced by the difference in state-of-the-art (SOTA) for area under the receiver operating characteristic curve (AUROC). For instance, for a model trained on CIFAR-100 (which consists of classes such as mammals, fish, flowers, fruits, household devices, trees, vehicles, insects, etc), a far-OOD task would be detecting digits from the street-view house numbers (SVHN) dataset as outliers. For the same model, detecting images from the CIFAR-10 dataset (which consists of the following 10 classes: airplane, automobile, bird, cat, deer, dog, frog, horse, ship, truck) would be considered a near-OOD task, which is more difficult as the classes are semantically similar. There has been impressive progress on far-OOD detection, for instance there are several approaches which can achieve AUROC close to 99% on CIFAR-100 (in) vs SVHN (out) task, cf. [Sastry and Oore, 2020]. However, the state-of-the-art for near-OOD detection is much lower, for instance the SOTA AUROC for CIFAR-100 (in) vs CIFAR-10 (out) task is around 85%

---

[*]Equal contribution.

35th Conference on Neural Information Processing Systems (NeurIPS 2021).

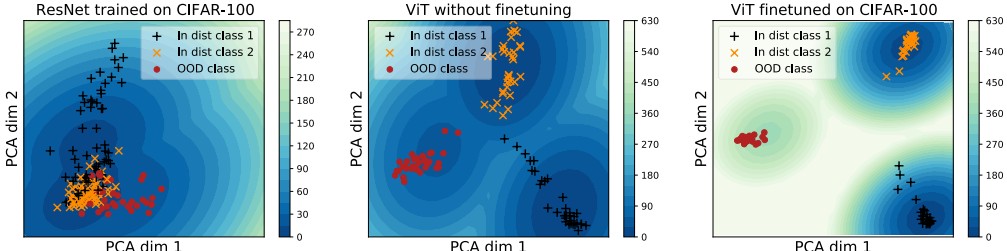

Figure 1: A two-dimensional PCA projection of the space of embedding vectors for 3 models, with examples of 2 in-distribution (from CIFAR-100) and 1 out-of-distribution class (from CIFAR-10). The color coding shows the Mahalanobis outlier score, while the points are projections of embeddings of members of the in-distribution CIFAR-100 classes "sunflowers" (black plus signs) and "turtle" (yellow crosses), and the OOD CIFAR-10 class "automobile" (red circles). The left panel shows a ResNet-20 trained on CIFAR-100, which assigns low Mahalanobis distance to OOD inputs and leads to overlapping clusters of class embeddings. The ViT pre-trained on ImageNet-21k (middle panel) is able to distinguish classes from each other well, but does not lead to well-separated outlier scores. ViT fine-tuned on CIFAR-100 (right panel) is great at clustering embeddings based on class, as well as assigning high Mahalanobis distance to OOD inputs (red).

[Zhang et al., 2020] which is considerably lower than the SOTA for far-OOD tasks. Similar trends are observed in other modalities such as genomics where the SOTA AUROC of near-OOD detection is only 66% [Ren et al., 2019]. Improving the SOTA for these near-OOD detection tasks and closing the performance gap between near-OOD detection and far-OOD detection is one of the key challenges in ensuring the safe deployment of models.

Large-scale pre-trained transformers have led to significant accuracy improvements in multiple domains, cf. Bidirectional Encoder Representations from Transformers (BERT) for text [Devlin et al., 2018], Vision Transformers (ViT) for images [Dosovitskiy et al., 2021], Contrastive Language–Image Pre-training (CLIP) trained on image-text pairs [Radford et al., 2021]. We show that classifiers obtained by fine-tuning large-scale pre-trained transformers are significantly better at near-OOD detection. Intuitively, large-scale pre-training makes classifiers less vulnerable to *shortcut learning* [Geirhos et al., 2020], making these representations better suited for near-OOD detection. Figure 1 visualizes two-dimensional PCA projections of representations from residual networks (ResNet) [He et al., 2016] trained on CIFAR-100 and ViT model pre-trained on ImageNet-21k and fine-tuned on CIFAR-100; we can observe that representations obtained by fine-tuning pre-trained transformers are better suited at near-OOD detection than representations from ResNet just trained on CIFAR-100.

Motivated by real-world applications which demand very high level of OOD detection for safe deployment, we explore variants of outlier exposure to further improve OOD detection. We show that pre-trained transformers are particularly well-suited at leveraging known outliers due to their high-quality representations (see Figure 1). We systematically vary the number of outlier examples per class, and show that even a handful of known outliers can significantly improve OOD detection. We refer to this setting as *few-shot outlier exposure*. For multi-modal pre-trained transformers, we explore a new form of outlier exposure that leverages names of outlier classes without any accompanying images, and show that this can significantly improve OOD detection for zero-shot classification.

In summary, our contributions are the following:

- We show that pre-trained transformers lead to significant improvements on near-OOD benchmarks. Concretely, we improve the AUROC of OOD detection on CIFAR-100 vs CIFAR-10 from 85% (current SOTA) to 96% using ViT pre-trained on ImageNet-21k, and improve the AUROC on a genomics OOD detection benchmark from 66% (current SOTA) to 77% using BERT.

- We show that pre-trained transformers are well-suited for few-shot outlier exposure. With just 10 labeled examples per class, we can improve the AUROC of OOD detection on CIFAR-100 vs CIFAR-10 to 99%, and improve the AUROC of OOD detection on genomics to 86%.

- We explore OOD detection for pre-trained multi-modal image-text transformers in the zero-shot classification setting, and show that just using the names of outlier classes as candidate text labels for CLIP, we can achieve AUROC of 94.8% on CIFAR-100 vs CIFAR-10 task. On easier far-OOD tasks such as CIFAR-{100, 10} vs SVHN, we achieve AUROC of 99.6% and 99.9% respectively.

## 2 Background and Related work

**Notation** We assume that we have an in-distribution dataset $\mathcal{D}^{\text{in}}$ of $(\boldsymbol{x}^{\text{in}}, y^{\text{in}})$ pairs where $\boldsymbol{x}$ denotes the input feature vector, and $y^{\text{in}} \in \mathcal{Y}^{\text{in}} := \{1, \ldots, K\}$ denotes the class label. Let $\mathcal{D}^{\text{out}}$ denote an out-of-distribution dataset of $(\boldsymbol{x}^{\text{out}}, y^{\text{out}})$ pairs where $y^{\text{out}} \in \mathcal{Y}^{\text{out}} := \{K+1, \ldots, K+O\}, \mathcal{Y}^{\text{out}} \cap \mathcal{Y}^{\text{in}} = \emptyset$. Depending on how different $\mathcal{D}^{\text{out}}$ is from $\mathcal{D}^{\text{in}}$, we categorize the OOD detection tasks into near-OOD and far-OOD. We first study the scenario where the model is fine-tuned only on the training set $\mathcal{D}^{\text{in}}_{\text{train}}$ without any access to OOD data. The test set contains $\mathcal{D}^{\text{in}}_{\text{test}}$ and $\mathcal{D}^{\text{out}}_{\text{test}}$ for evaluating OOD performance using AUROC. Next, we explore the scenario where a small number of OOD examples are available for training, i.e. the few-shot outlier exposure setting. In this setting, the training set contains $\mathcal{D}^{\text{in}}_{\text{train}} \bigcup \mathcal{D}^{\text{out}}_{\text{few-shot}}$, where $|\mathcal{D}^{\text{out}}_{\text{few-shot}}|$ is often smaller than 100 per OOD class.

### 2.1 Methods for detecting OOD inputs

We describe a few popular techniques for detecting OOD inputs using neural networks.

**Maximum over softmax probabilities (MSP)** A baseline method for OOD detection is to use the maximum softmax probability as the confidence score, i.e. $\text{score}_{\text{msp}}(\boldsymbol{x}) = \max_{c=1,\ldots,K} p(y = c | \boldsymbol{x})$ [Hendrycks and Gimpel, 2016]. While being slightly worse than other techniques, its simplicity and performance make it an ideal baseline.

**Mahalanobis distance** Lee et al. [2018] proposed to fit a Gaussian distribution to the class-conditional embeddings and use the Mahalanobis distance for OOD detection. Let $f(\boldsymbol{x})$ denote the embedding (e.g. the penultimate layer before computing the logits) of an input $\boldsymbol{x}$. We fit a Gaussian distribution to the embeddings of the training data, computing per-class mean $\boldsymbol{\mu}_c = \frac{1}{N_c} \sum_{i:y_i=c} f(\boldsymbol{x}_i)$ and a shared covariance matrix $\boldsymbol{\Sigma} = \frac{1}{N} \sum_{c=1}^{K} \sum_{i:y_i=c} (f(\boldsymbol{x}_i) - \boldsymbol{\mu}_c)(f(\boldsymbol{x}_i) - \boldsymbol{\mu}_c)^{\top}$. The Mahalanobis score (negative of the distance) is then computed as: $\text{score}_{\text{Maha}}(\boldsymbol{x}) = -\min_c \left( \frac{1}{2}(f(\boldsymbol{x}) - \boldsymbol{\mu}_c) \boldsymbol{\Sigma}^{-1} (f(\boldsymbol{x}) - \boldsymbol{\mu}_c)^{\top} \right)$.

**Outlier exposure** Hendrycks et al. [2018] proposed *outlier exposure* which leverages a large dataset of known outliers. For classification problems, the model is trained to predict uniform distribution over labels for these inputs. Thulasidasan et al. [2021] proposed to use a single outlier class as the $(K+1)^{\text{th}}$ class for a $(K+1)$-way classification problem. Roy et al. [2021] showed that leveraging the labels of known outliers (rather than assigning all known outliers to a single class) can further improve OOD detection performance.

### 2.2 Pre-training neural networks

Architectures using self-attention, typically based on the Transformer [Vaswani et al., 2017], often combined with large-scale pre-training directly on raw text, have been very popular for natural language processing (NLP) tasks in recent years [Devlin et al., 2018, Dai and Le, 2015, Peters et al., 2018, Howard and Ruder, 2018, Radford et al., 2018, Raffel et al., 2019]. Often followed by fine-tuning on a smaller, downstream dataset, large-scale pre-training techniques lead to highly informative embeddings that are broadly useful in natural language tasks. The advantage of the transformer architecture is its ability to scale to very large model sizes, reaching up to 1 trillion parameter mark [Fedus et al., 2021]. Large pre-trained models, such as GPT-3 [Brown et al., 2020], have shown the potential of large-scale task-agnostic pre-training in language.

The Vision Transformer (ViT) [Dosovitskiy et al., 2021] has shown that self-attention combined with large-scale pre-training is a viable strategy for vision tasks as well. The performance of ViT is comparable to other state-of-the-art models, while being more efficient to train. Its ability to quickly fine-tune to a smaller, downstream dataset, generalizing even in a few-shot regime, makes it an attractive backbone for tasks such as out-of-distribution (OOD) detection. In this paper, we use ViT pre-trained on ImageNet-21k [Ridnik et al., 2021].

Multi-modal text-image transformers such as CLIP (Contrastive Language-Image Pre-Training) [Radford et al., 2021] pre-train on 400 million (image, text) pairs from the internet to learn to predict a raw text caption from an image, and by doing so develop state-of-the-art visual representations and their natural language counterparts. Radford et al. [2021] showed that CLIP improves robustness to natural distribution shift. In this paper, we use the shared image-text embedding to introduce a new zero-shot OOD detection method (Section 5). Hendrycks et al. [2019a] show that pre-training improves OOD detection for non-transformer architectures. Self-supervised learning techniques have

also been shown to improve OOD detection; Hendrycks et al. [2019b] use rotation prediction and Winkens et al. [2020] use contrastive training to improve near-OOD detection.

**Robustness of pre-trained transformers** Hendrycks et al. [2020] show that pre-trained transformers improve OOD detection in NLP. Pre-trained BERT has been has used as a backbone for OOD detection in language, cf. [Liu et al., 2020]. Unlike them, we focus on vision and genomics modalities, and specifically on near-OOD detection benchmarks. Investigating the robustness of pre-trained ViT is an active research area and there are several concurrent papers exploring robustness of ViT to adversarial perturbations and common corruptions (ImageNet-C). Bhojanapalli et al. [2021] show the robustness of pre-trained transformers to input perturbations, and of transformers to layer removal. Caron et al. [2021] demonstrate many emerging properties in self-supervised ViTs, while Shao et al. [2021], Mahmood et al. [2021] show that they are more robust to adversarial perturbations, and Naseer et al. [2021] that they have less texture bias. Paul and Chen [2021], Minderer et al. [2021] show that ViT is more robust to distribution shift and natural adversarial examples [Paul and Chen, 2021], and Mao et al. [2021] propose a robust ViT. To the best of our knowledge, we are the first to show that pre-trained ViT can significantly improve near-OOD detection in vision benchmarks, and show that few-shot outlier exposure can further improve performance.

## 3 Near-OOD detection on image classification benchmarks

### 3.1 Fine-tuning the Vision Transformer

We use the Vision Transformer (ViT) architecture [Dosovitskiy et al., 2021] and its pre-trained model checkpoints.[2] The checkpoints are pre-trained on ImageNet-21k [Deng et al., 2009]. We fine-tune the full ViT architecture on a downstream task that is either the CIFAR-10 or CIFAR-100 classification problem (using a TPU in Google Colab). Once the model is fine-tuned, we get its pre-logit embeddings (the layer immediately preceding the final layer) for the train and test sets of CIFAR-10 and CIFAR-100 to use for OOD tasks. We use the maximum over softmax probabilities (labeled as MSP) and the Mahalanobis distance (labeled as Maha).

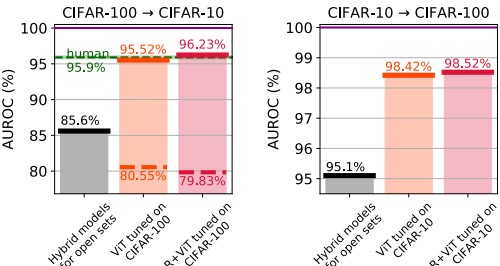

Figure 2: Left: CIFAR-100 vs CIFAR-10 OOD AUROC for previous state-of-the-art Zhang et al. [2020], our fine-tuned ViT with two different backbones (ViT-B_16 and R50+ViT-B_16). Right: CIFAR-10 vs CIFAR-100 OOD task.

Table 1: ImageNet-21k pre-trained ViT/BiT/MLP-Mixer fine-tuned on the in-distribution training set.

| Model | In-distribution | fine-tuned test accuracy | Out-distribution | Mahalanobis AUROC | MSP AUROC |
|---|---|---|---|---|---|
| BiT-M R50x1 | CIFAR-100 | 87.01% | CIFAR-10 | 81.71% | 81.15% |
| BiT-M R101x3 | CIFAR-100 | 91.55% | CIFAR-10 | 90.10% | 83.69% |
| ViT-B_16 | CIFAR-100 | 90.95% | CIFAR-10 | 95.53% | 91.89% |
| R50+ViT-B_16 | CIFAR-100 | 91.71% | CIFAR-10 | **96.23**% | 92.08% |
| MLP-Mixer-B_16 | CIFAR-100 | 90.40% | CIFAR-10 | 95.31% | 90.22% |
| BiT-M R50x1 | CIFAR-10 | 97.47% | CIFAR-100 | 95.52% | 85.87% |
| BiT-M R101x3 | CIFAR-10 | 97.36% | CIFAR-100 | 94.55% | 85.34% |
| ViT-B_16 | CIFAR-10 | 98.10% | CIFAR-100 | 98.42% | 97.68% |
| R50+ViT-B_16 | CIFAR-10 | 98.70% | CIFAR-100 | **98.52**% | 97.75% |
| MLP-Mixer-B_16 | CIFAR-10 | 97.58% | CIFAR-100 | 97.85% | 96.28% |

The results are summarized in Table 1 and Figure 2. We observe that the MSP baseline yields surprisingly good results when used on top of a large pre-trained transformer that has been fine-tuned on the in-distribution training set. The Mahalanobis distance technique improves OOD detection even further. Applying Mahalanobis distance to a pre-trained ViT fine-tuned on CIFAR-100, we achieve AUROC of 96% on CIFAR-100 vs CIFAR-10, significantly improving over the previous SOTA of

---

[2] https://github.com/google-research/vision_transformer

85% using a hybrid model [Zhang et al., 2020]. To study the effect of model architecture, we also evaluate OOD performance on another large-scale pre-trained model, Big Transfer (BiT) [Kolesnikov et al., 2019][3], as a comparison to ViT. We use the BiT-M R50x1 and R101x3 models pre-trained on ImageNet-21k, and fine-tune the full model architecture on CIFAR-10 and CIFAR-100 respectively. The results are shown in Table 1. For both directions, the AUROCs for BiT are lower than that for ViT. More importantly, BiT uses a different model architecture, ResNet, instead of a transformer, which may explains the large difference in the OOD performance. As an additional ablation, we fine-tuned the MLP-Mixer pre-trained on ImageNet-21k [Tolstikhin et al., 2021][4], a high-performance all-MLP architecture for vision, and compared its performance to the Vision Transformer (ViT) and BiT. The summary of our results can be found in Table 1. We observe that MLP-mixer outperforms BiT as well, which adds additional evidence that pre-training helps architectures such as ViT and MLP-mixer more than it helps BiT.

Due to semantic similarity between classes in CIFAR, this task is hard for humans as well.[5] We also evaluated the performance of our approach on popular far-OOD benchmarks such as CIFAR-* vs SVHN and CIFAR-* vs Textures, and achieve very high AUROC values of around 98% or higher, see Table 7 in Appendix C. We mainly focus on the difficult near-OOD as this is a more challenging and realistic problem; many methods can achieve high AUROC on the easier far-OOD benchmarks, but do not perform as well in near-OOD tasks, cf. [Winkens et al., 2020, Table 1] which compares many methods on near-OOD and far-OOD tasks.

Since ViT models are typically pre-trained using a large labeled set, we ran additional ablations to understand how much of the improvement is due to supervision vs transformers. To assess the role of supervised pre-training, we compared the results to a ViT pre-trained on ImageNet-21k in a self-supervised way that does not use any labels. We took a pre-trained checkpoint from Caron et al. [2021], and fine-tuned it on CIFAR-100. The results are shown as DINO ViT-B_16 in Table 2. Since the fine-tuned test accuracy is lower for DINO ViT-B_16, we also include an ViT-B_16 that was fine-tuned for fewer steps to achieve comparable test accuracy as DINO ViT-B_16. Note that even though DINO ViT-B_16 is pre-trained without labels, the AUROC is significantly higher than the current SOTA for CIFAR-100 vs CIFAR-10. The difference between DINO ViT-B_16 vs early stopped ViT-B_16 shows the difference due to supervision during pre-training. We also experimented with larger ViT models such as ViT-L_16 and ensembles, and found that they improve AUROC of OOD detection to 98% on CIFAR-100 vs CIFAR-10 task. We found that the OOD detection is lower for ViT models with lower fine-tuned test accuracy, see Appendix C.1 for these results. Hence, we believe that better strategies for unsupervised pre-training and fine-tuning can further improve OOD detection performance. In Section 4, we explore unsupervised pre-training for transformers to improve near-OOD detection in genomics.

Table 2: Additional ablations to measure the effect of supervised pre-training. * indicates self-supervised pre-training without labels.

| Model | In-distribution | fine-tuned test accuracy | Out-distribution | Mahalanobis AUROC | MSP AUROC |
|---|---|---|---|---|---|
| DINO ViT-B_16* | CIFAR-100 | 88.95% | CIFAR-10 | 88.78% | 81.25% |
| ViT-B_16 (early stop) | CIFAR-100 | 88.71% | CIFAR-10 | 93.05% | 88.82% |
| ViT-B_16 | CIFAR-100 | 90.95% | CIFAR-10 | 95.53% | 91.89% |
| R50+ViT-B_16 | CIFAR-100 | 91.71% | CIFAR-10 | **96.23%** | 92.08% |
| ViT-L_16 | CIFAR-100 | 94.73% | CIFAR-10 | **97.98%** | 94.28% |
| ViT ensemble | CIFAR-100 | — | CIFAR-10 | **98.11%** | 95.15% |

---

[3] https://github.com/google-research/big_transfer

[4] https://github.com/google-research/vision_transformer

[5] To approximately estimate the human performance, we performed the CIFAR-100 vs CIFAR-10 near-OOD detection ourselves. We set-up a simple GUI to randomly sample images from both CIFAR-100 and CIFAR-10 datasets, where the task was to identify images that belong to the CIFAR-10 classes. Note that this setup is easier for humans as they only have to remember 10 classes in CIFAR-10 (as opposed to the 100 classes in CIFAR-100). The estimated human performance was 96% AUROC which demonstrates the difficulty of the task. We do not claim that this is the best possible human performance, we will open source the code so that readers can estimate their OOD detection performance themselves. Further details are available in Appendix A.

## 3.2 Few-shot outlier exposure using ViT

In Section 3.1, we demonstrated that fine-tuning a pre-trained ViT model can improve near-OOD detection (with relatively simple OOD detection techniques such as MSP and Mahalanobis distance). Figure 1 shows that the representations from fine-tuned ViT models are well-clustered. This motivates *few-shot outlier exposure*, where we assume just a handful of known outliers (and optionally their labels). This setting is motivated by real-world applications which require high-quality OOD detection and teams are willing to collect a handful of known outlier examples (rather than just rely on modeling approaches) to improve safety. Another setting is the case where models are being continuously re-trained; once an outlier is detected, it is desirable to include that in the training corpus to encourage the model to correctly detect similar examples as outliers in the future.

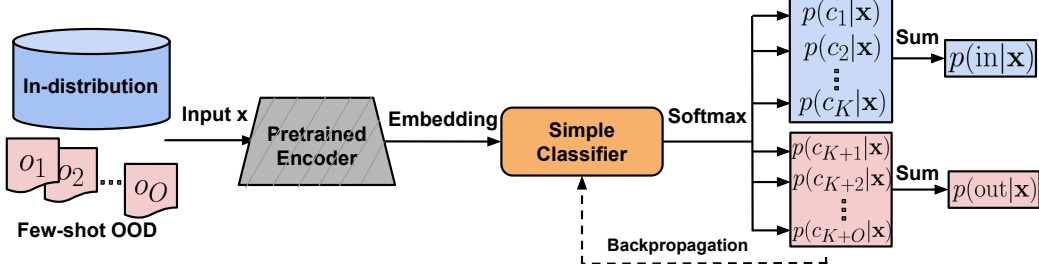

Figure 3: Few-shot outlier exposure with pre-trained transformers. The OOD samples are used to fine-tune a simple classifier (linear classifier for ViT which uses supervised pre-training, and shallow MLP with one hidden layer for genomics which uses unsupervised pre-training). We use the in-distribution classes in addition to multiple OOD classes (when labels available), or a single OOD class. The confidence score is the sum of probabilities corresponding to the in-distribution classes.

The general approach is shown in Figure 3. By using the in-distribution training set $D_{\text{train}}^{\text{in}}$ with $K$ classes and a small number of known OOD examples from $D_{\text{few-shot}}^{\text{out}}$ with $O$ classes, we train a simple classifier $h(\cdot)$ that maps an embedding vector $\boldsymbol{z}$ to a probability vector $\boldsymbol{p} \in \mathbb{R}^{K+O}$, which concatenates the in- and out-of-distribution classes. We considered two types of outlier exposure, one which assumes access to outlier labels (similar to [Roy et al., 2021]) and one where all the outlier examples are collapsed to a single $(K+1)$th class (similar to [Thulasidasan et al., 2021]). For models pre-trained with labels (such as ViT), we use a linear classifier. For models that use unsupervised pre-training (e.g. genomics in Section 4), we use a shallow multi-layer perceptron (MLP) with a single hidden layer so that fine-tuning can learn discriminative features. We use the sum of the probabilities of all $K$ in-distribution classes as the confidence score for the OOD detection task, $\text{score}_{\text{oe}}(\boldsymbol{x}) = p(\text{in}|\boldsymbol{x}) = \sum_{c=1,...,K} p(y = c|\boldsymbol{x})$. When training the MLP $h(\cdot)$ with few-shot OOD examples, there could be many more examples of the in-distribution data than the small number of OOD data. We therefore oversample the OOD inputs by a factor that we calculate as $(|\mathcal{D}_{\text{train}}^{\text{in}}|/|\mathcal{D}_{\text{oe}}^{\text{out}}|)(O/K)$. This makes the training classes approximately balanced during training. We used a single layer MLP from scikit-learn [Pedregosa et al., 2011], batch size 200, $L_2$ penalty of 1, learning rate 0.001 with *Adam*, maximum of 1,000 iterations. Algorithm 1 and Algorithm 2 describe the details of training and scoring.

---

**Algorithm 1** Few-shot outlier exposure training

1: **Input:** In-distribution train set $\mathcal{D}_{\text{train}}^{\text{in}} = \{(\boldsymbol{x}, y)\}$ with $K$ classes, out-of-distribution train subset $\mathcal{D}_{\text{few-shot}}^{\text{out}} = \{(\boldsymbol{x}, y)\}$ with $O$ classes, oversampling factor $\Gamma$, a pretrained feature extractor $f(\cdot) : \boldsymbol{x} \to \boldsymbol{z}$, a simple classification head $h(\cdot) : \boldsymbol{z} \to \boldsymbol{p} \in \mathbb{R}^{K+O}$.
2: **Initialize:** Initialize $h(\cdot)$ at random, generate random batches from $\mathcal{D}_{\text{train}}^{\text{in}} \bigcup \mathcal{D}_{\text{train}}^{\text{out}}$, oversampling $\mathcal{D}_{\text{train}}^{\text{out}}$ by $\Gamma$.
3: **for** train_step = 1 **to** max_step **do**
4:     loss = CrossEntropy$(h(f(\boldsymbol{x})), y)$
5:     SGD update of $h(\cdot)$ w.r.t loss
6: **end for**

**Algorithm 2** Few-shot outlier inference

1: **Input:**
In-distribution test set $\mathcal{D}_{\text{test}}^{\text{in}} = \{(\boldsymbol{x}, y)\}$ with $K$ classes, out-of-distribution test subset $\mathcal{D}_{\text{test}}^{\text{out}} = \{(X, y)\}$ with $O$ classes, a pre-trained $f(\cdot) : \boldsymbol{x} \to \boldsymbol{z}$ from inputs to embedding vectors, a trained classification head $h(\cdot) : \boldsymbol{z} \to \mathbf{p} \in \mathbb{R}^{K+O}$.
2: Compute $\text{score}_{\text{oe}}^{\text{in}}(\boldsymbol{x}), \boldsymbol{x} \in \mathcal{D}_{\text{test}}^{\text{in}}$
3: Compute $\text{score}_{\text{oe}}^{\text{out}}(\boldsymbol{x}), \boldsymbol{x} \in \mathcal{D}_{\text{test}}^{\text{out}}$
4: Compute AUROC based on the scores.

---

We systemically vary the number of known outliers from 1-10 examples per OOD class and evaluate the OOD detection performance. We also show results for higher numbers of examples per class for

completeness as it illustrates how quickly the performance saturates. Figure 4 and Table 3 show the few-shot outlier exposure results for CIFAR-100 vs CIFAR-10 and CIFAR-10 vs CIFAR-100. We evaluate performance of the pre-trained transformer (without any fine-tuning on CIFAR-*) as well as a fine-tuned transformer. We observe that even with 1-10 known outliers per class, we can achieve 99% AUROC for near-OOD detection on CIFAR-100 vs CIFAR-10. Interestingly, we observe that having labels for outliers is less important when the transformer is fine-tuned on in-distribution (dashed vs solid red lines) than in the scenario where the transformer is not fine-tuned (dashed vs solid blue lines). Intuitively, the embeddings obtained by fine-tuning a pre-trained transformer are well-clustered, so just a handful of known outliers can significantly improve OOD detection, as illustrated in Figure 9 in Appendix B.

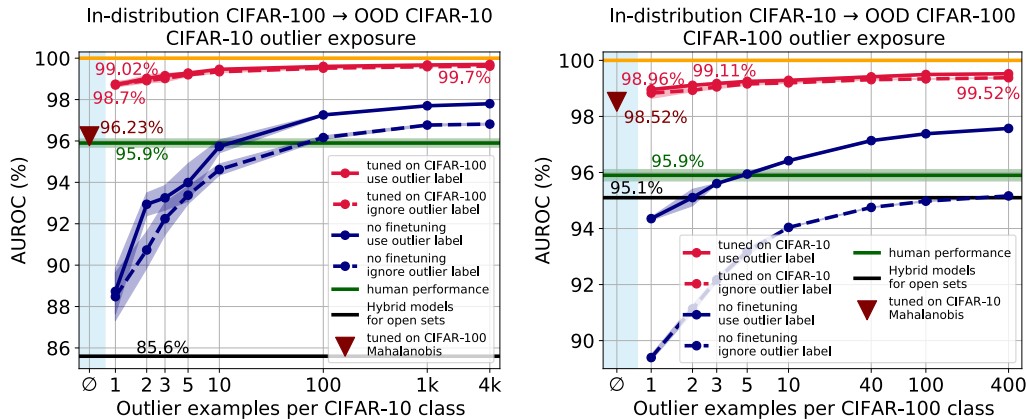

Figure 4: The effect of few-shot outlier exposure and fine-tuning on CIFAR-100 vs CIFAR-10 (left) and CIFAR-10 vs CIFAR-100 (right) using a R50+ViT-B_16 pre-trained on ImageNet-21k. Fine-tuning on in-distribution (red) prior to outlier exposure outperforms no fine-tuning (blue).

Table 3: ImageNet-21k pre-trained ViT (optionally fine-tuned on in-distribution), with an additional final layer that was trained using the in-distribution train set and a small number of examples of the OOD train set (including the $O$ OOD class labels, corresponding to the solid curves in Figure 4).

(a) CIFAR-100 vs CIFAR-10 AUROC results.

| Number of OOD (CIFAR-10) examples per class | R+ViT (without fine-tuning) | R+ViT fine-tuned on CIFAR-100 |
|---|---|---|
| 1 | $88.73 \pm 1.08\%$ | $98.70 \pm 0.08\%$ |
| 2 | $92.94 \pm 0.55\%$ | $99.02 \pm 0.15\%$ |
| 3 | $93.25 \pm 0.59\%$ | $99.16 \pm 0.11\%$ |
| 10 | $95.73 \pm 0.31\%$ | $99.46 \pm 0.01\%$ |
| 100 | $97.70 \pm 0.01\%$ | $99.67 \pm 0.01\%$ |

(b) CIFAR-10 vs CIFAR-100 AUROC results.

| Number of OOD (CIFAR-100) examples per class | R+ViT (without fine-tuning) | R+ViT fine-tuned on CIFAR-10 |
|---|---|---|
| 1 | $94.35 \pm 0.05\%$ | $98.96 \pm 0.05\%$ |
| 2 | $95.10 \pm 0.30\%$ | $99.11 \pm 0.04\%$ |
| 3 | $95.60 \pm 0.01\%$ | $99.17 \pm 0.03\%$ |
| 10 | $96.42 \pm 0.02\%$ | $99.29 \pm 0.02\%$ |
| 100 | $97.38 \pm 0.01\%$ | $99.50 \pm 0.01\%$ |

# 4   Near OOD detection of genomic sequences

We investigate OOD detection in genomics as another input modality for near-OOD detection. Ren et al. [2019] proposed a benchmark dataset[6] for OOD detection in genomics, motivated by the real-world problem of bacteria identification based on genomic sequences. Real bacteria sequencing data can contain approximately 60-80% of sequences from unknown classes that have not been studied before. Hence, a classifier trained on all known classes so far will be inevitably asked to predict on genomes that do not belong to one of the known classes. Since different bacteria classes are discovered gradually over the years, Ren et al. [2019] use a set of 10 bacteria classes that were discovered before the year 2011 as in-distribution classes, a set of 60 bacteria classes discovered between 2011-2016 as the validation OOD, and a set of 60 different bacteria classes discovered after 2016 as the test OOD. The training set only contains genomic sequences of in-distribution classes. The validation and test sets contain sequences from both in-distribution and OOD classes. The genomic sequence is of fixed length of 250 base pairs, composed by characters of A, C, G, T. In

---

[6]https://www.tensorflow.org/datasets/catalog/genomics_ood

the previous work, 1-dimensional Convolutional Neural Networks (1D CNN) were used to build the classifier for the 10 in-distributional classes, and the maximum of softmax probabilities (MSP) and Mahalanobis distance were used for OOD detection. The best AUROC for MSP was only 66.14%, and 62.41% for Mahalanobis distance [Ren et al., 2019].

Similar to the previous section, we explore the usefulness of pre-trained transformers and fine-tuning for near-OOD detection. Unsupervised pre-training and fine-tuning approach has been applied to several bio-informatic problems, such as protein function prediction [Elnaggar et al., 2020, Dohan et al., 2021, Littmann et al., 2021], protein structure prediction [Rives et al., 2021], and predicting promoters, splice sites and transcription factor binding sites [Ji et al., 2020], but it has not yet been studied how pre-training could help near-OOD detection.

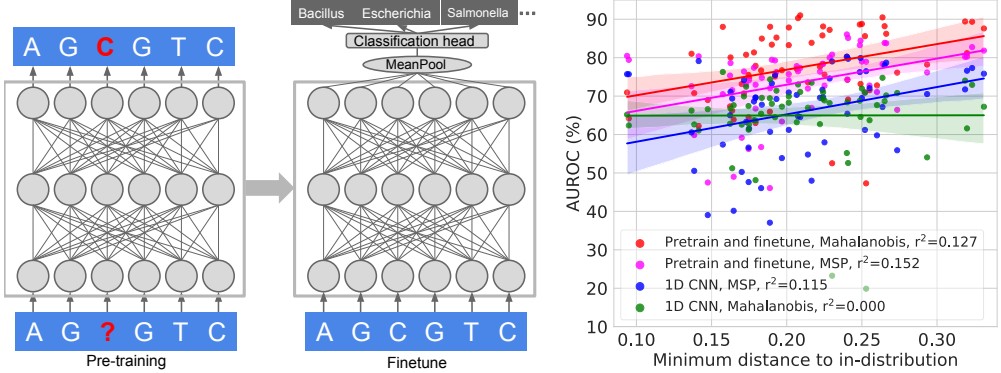

(a) BERT pre-training and fine-tuning for genomics.        (b) AUROC of near-OOD detection

Figure 5: (a) Model architecture for BERT pre-training and fine-tuning. The unsupervised pre-training model uses a transformer encoder to predict the masked token (shown in red). The fine-tuned model adds a simple classification head (a single linear projection) to predict in-distribution classes. (b) The relationship between the minimum genetic distance and the AUROC scores for the 60 OOD test classes. The pre-train+fine-tune based MSP and Mahalanobis distance methods have significantly higher AUROC overall, and the positive correlation between the minimum distance and the AUROC are more prominent than for the baseline model.

**Unsupervised BERT pre-training and supervised fine-tuning**   We first pre-train the transformer model in an unsupervised fashion as in BERT to capture biologically relevant properties, following Dohan et al. [2021]. For unlabeled in-distribution sequences in the training set, we randomly mask the characters in the sequence at the rate of 0.15, feed the masked sequence into transformer-based model of 8 heads and 6 layers and embedding dimension 512, and predict the masked characters. To boost the performance, we add the unlabeled validation data to the training set for pre-training. In the fine-tuning stage, we load the pre-trained transformer model, mean pool the embeddings over the positions, and add a single linear projection classification head for 10 in-distribution classes on top of the embeddings. The setup is shown in Figure 5a. All the parameters in the model including those in the pre-trained transformer and those in the classification head are fine-tuned using the labeled training data. The model is pre-trained for 300,000 steps using learning rate of 0.001 and Adam optimizer [Kingma and Ba, 2014] on TPU, and the accuracy for predicting the masked token is 48.35%. The model is fine-tuned for 100,000 steps at the learning rate of 0.0001, and the classification accuracy is 89.84%. We use the validation in-distribution and validation OOD data to select the best model checkpoint for each of the two methods and evaluate on test set.

Table 4: Genomics OOD BERT pre-trained and fine-tuned on the in-distribution training set. Error bars represent standard deviation over 3 runs. See Table 11 in Appendix D for AUPRC and FPR95.

| Model | Test Accuracy | Mahalanobis AUROC | MSP AUROC |
|---|---|---|---|
| 1D CNN [Ren et al., 2019] | 85.93±0.11% | 64.75±0.73% | 65.84±0.46% |
| BERT pre-train and fine-tune | 89.84±0.00% | **77.49±0.04%** | 73.53±0.03% |

The results are reported in Table 4. It can be seen that using the approach of pre-training transformer and fine-tuning, the OOD detection performance is significantly improved, from 64.75% to 77.49%

for Mahalanobis distance, and from 65.84% to 73.53% for MSP. The in-distribution accuracy also improves a bit, from 85.93% to 89.84%. We also study the relationship between the genetic distance and the AUROC of OOD detection for the 60 test OOD classes. We compute the genetic distance using the popular alignment-free method $d_2^S$ which is based on the similarity between the word frequencies of the two genomes [Ren et al., 2018, Reinert et al., 2009]. Studies have shown that this genetic distance reflects true evolutionary distances [Chan et al., 2014, Bernard et al., 2016]. For each of the 60 OOD test classes, we use the minimum genetic distance between this OOD class to any of the 10 in-distribution classes as the final distance measure. Figure 5 shows the AUROC and the minimum distance for each of the 60 OOD classes. We expect the AUROC is higher as the distance is greater. Using the baseline 1D CNN model, we did not see obvious correlation between the AUROC and the minimum distance, with $r^2 = 0.0000$, based on Mahalanobis distance. The AUROC based on MSP method has positive correlation to the minimum distance, with $r^2 = 0.1190$. After we use the pre-trained+fine-tuned transformer, both MSP and Mahalanobis distance methods have significantly higher AUROC overall, and the positive correlation between the minimum distance and the AUROC is more prominent than for the baseline model.

**Few-shot outlier exposure** Given that pre-trained and fine-tuned model improves the OOD performance, we next explore the idea of few shot outlier exposure to further boost the performance. We randomly select 1, 2, 5, 10, 100 examples per test OOD class and add them to the training set respectively. For each input $x$ in the training set, we extract its corresponding embedding vector $z$ from the above pre-trained and fine-tuned model (or alternatively the model without fine-tuning). We construct a single layer perceptron network of 1024 units for classifying each individual to in-distribution classes and OOD classes, as shown in Figure 3. At inference time, we use the sum of the probability of in-distribution classes as the final confidence score for OOD detection. Additionally, we also tried the idea of collapsing all OOD classes into one single class (as in [Thulasidasan et al., 2021]) for comparison. The model is

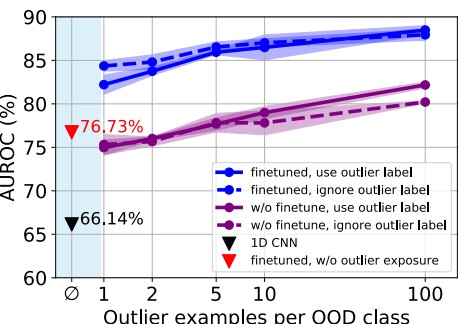

Figure 6: Few-shot outlier exposure for genomics OOD. The x-axis shows the number of outliers per class that the model was exposed to. The y-axis is OOD AUROC in %. The shading shows the standard deviation over 3 runs. See Table 12 for exact numbers.

trained for 10,000 steps with the learning rate of 0.001. The best model checkpoint is selected based on the highest AUROC on a small set of validation dataset disjoint from the test set.

Results are shown in Figure 6. We observe that exposing to just a small number of OOD examples significantly improves the OOD performance, increasing AUROC from 76.73% to 88.48%. As expected, using the embeddings from the fine-tuned model (blue lines) is better than that from the model without fine-tuning (purple lines). Also, using the outlier labels (purple solid line) has a slightly better performance than collapsing the OOD classes into a single class (purple dashed line) using the pre-trained embeddings without fine-tuning.

## 5 Using candidate labels with multi-modal text-image models such as CLIP

Multi-modal transformers such as CLIP [Radford et al., 2021], which are pre-trained on image-text pairs, have been shown to perform well on zero-shot classification tasks. *We show that such multi-modal transformers open the door to new forms of outlier exposure which can significantly improve out-of-distribution (OOD) detection in the zero-shot classification setting.* Our goal is to show that multi-modal transformers can leverage a weaker form of outlier exposure than the few-shot outlier exposure assumption in previous sections, and improve their safety for zero-shot classification.

We use the pre-trained CLIP model[7] (specifically ViT-B/32) that was trained on 400 million (text, image) pairs from the internet. Its image encoder can map an image $I$ into an embedding vector $z_{\text{image}}(I)$, while its text encoder can do the same for a string $T$ as $z_{\text{text}}(T)$. By choosing a set of $D$ *candidate labels* for an image, the similarity between the embedding of the candidate label $T_i$ and an image $I$ can be used as the $i^{\text{th}}$ component of the image's embedding vector $z$ as $z_i = z_{\text{text}}(T_i) \cdot z_{\text{image}}(I)$.

---

[7]https://github.com/openai/CLIP

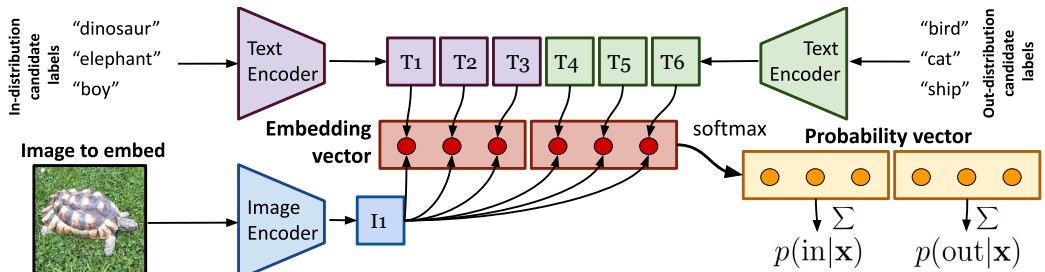

Figure 7: Using candidate text labels and an image-text multi-modal model (CLIP) to produce an embedding vector for OOD detection. We use two sets of candidate labels, evaluate the semantic alignment of the image with each label, apply softmax, and use the sum of probabilities in the first (in) set as an OOD score. Note that this is zero-shot classification and the model is not fine-tuned and does not leverage any in-distribution or OOD images/labels. It only uses the names of the classes (or other informative words) as candidate labels and works well due to the strong pre-training of CLIP.

**Zero-shot outlier exposure** In the zero-shot classification setting, the candidate labels are chosen to describe the semantic content of the in-distribution classes (e.g. names of the classes). We propose to include the candidate labels related to the out-of-distribution classes, and utilize this knowledge as a very weak form of outlier exposure in multi-modal models. This could be relevant in applications, where we might not actually have any outlier images for fine-tuning but we might know the *names* or *descriptions* of outlier classes.

Our proposed procedure is shown in Figure 7. We choose two groups of candidate labels, in-distribution and out-of-distribution labels (e.g. CIFAR-100 and CIFAR-10 class names). We produce an embedding vector $z$ for each image $I$, apply softmax to get probabilities as $\mathbf{p} = \mathrm{softmax}(z)$. Those get split to $p(\mathrm{in}|x) = \sum_{i \in \mathrm{in}} \mathbf{p}_i$, and $p(\mathrm{out}|x) = \sum_{i \in \mathrm{out}} \mathbf{p}_i$, where $p(\mathrm{in}|x) + p(\mathrm{out}|x) = 1$. Similar to Figure 3, we use $\mathrm{score}_{\mathrm{oe}}(x) = p(\mathrm{in}|x)$ as the confidence score. By choosing the candidate labels to represent the in-distribution and OOD dataset we would like to distinguish (e.g. CIFAR-100 and 10), we can get a very informative score that leads to AUROC above previous SOTA, despite no exposure to the training set of the in-distribution (zero-shot). Our results are shown in Table 5, with additional results in Table 10 of Appendix C.2.

Table 5: Zero-shot OOD detection using image-text multi-modal models. We compare CLIP that uses only the names of in-distribution classes (baseline) and compare it to our proposed variant that uses just the names of out-of-distribution classes as candidate labels. Even in the zero-shot setting (without any fine-tuning on either in-distribution or OOD dataset), we outperform previous SOTA.

| Distribution 1 | Distribution 2 | Labels 1 | Labels 2 | AUROC |
|---|---|---|---|---|
| CIFAR-100 | CIFAR-10 | CIFAR-100 names | — | 69.49% |
| CIFAR-100 | CIFAR-10 | CIFAR-100 names | CIFAR-10 names | 94.68% |
| CIFAR-10 | CIFAR-100 | CIFAR-10 names | — | 89.17% |
| CIFAR-10 | CIFAR-100 | CIFAR-10 names | CIFAR-100 names | 94.68% |
| CIFAR-100 | SVHN | CIFAR-100 names | — | 93.05% |
| CIFAR-100 | SVHN | CIFAR-100 names | ["number"] | 99.67% |
| CIFAR-10 | SVHN | CIFAR-10 names | — | 96.90% |
| CIFAR-10 | SVHN | CIFAR-10 names | ["number"] | 99.95% |

## 6 Conclusion

We focus on the challenging problem of near-OOD detection. We show that fine-tuning large-scale pre-trained transformers and using few-shot outlier exposure can significantly improve the SOTA. On the CIFAR-100 vs CIFAR-10 visual OOD detection benchmark, we improve the SOTA AUROC from 85% to 96% (without outlier exposure) and 99% (with outlier exposure), essentially closing the gap between SOTA and the ideal performance. On a challenging genomics benchmark, we improve the SOTA from 66% to 77% using BERT (without outlier exposure) and 88% (with outlier exposure). We also show that multi-modal pre-trained transformers open the door to new, weaker forms of outlier exposure which only use names of OOD inputs; we apply this to CLIP and achieve AUROC of 94.7% in the zero-shot classification setting. We believe that our findings will be of interest to the research community (and inspire the creation of harder near-OOD benchmarks) as well as practitioners working on safety-critical applications.

## Acknowledgements

We thank Abhijit Guha Roy, Jim Winkens, Jeremiah Liu, Lucas Beyer and the anonymous reviewers for helpful feedback. We thank David Dohan and Andreea Gane for the helpful advice on BERT genomics model pre-training. We thank Basil Mustafa for providing the BiT model checkpoints. We thank Matthias Minderer for his helpful advice on ViT. We thank Winston Pouse for useful discussions on human performance.

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
