# A    Measuring human performance on CIFAR-100 vs CIFAR-10 OOD task

While the typical accuracy a human reaches is often known for classification tasks, there is a lack of such benchmark for near-OOD detection. We decided to measure human performance on the task of distinguishing CIFAR-100 and CIFAR-10. To do that, we wrote a simple graphical user interface (GUI) where a user is presented with a fixed number of images randomly chosen from the in-distribution and out-of-distribution test sets (CIFAR-10 and 100 in our case). The user then clicks on the images they believe belong to the in-distribution. To make this easier, we allow the user to choose the images belonging to the individual classes of the in-distribution. An example of our GUI is shown in Figure 8.

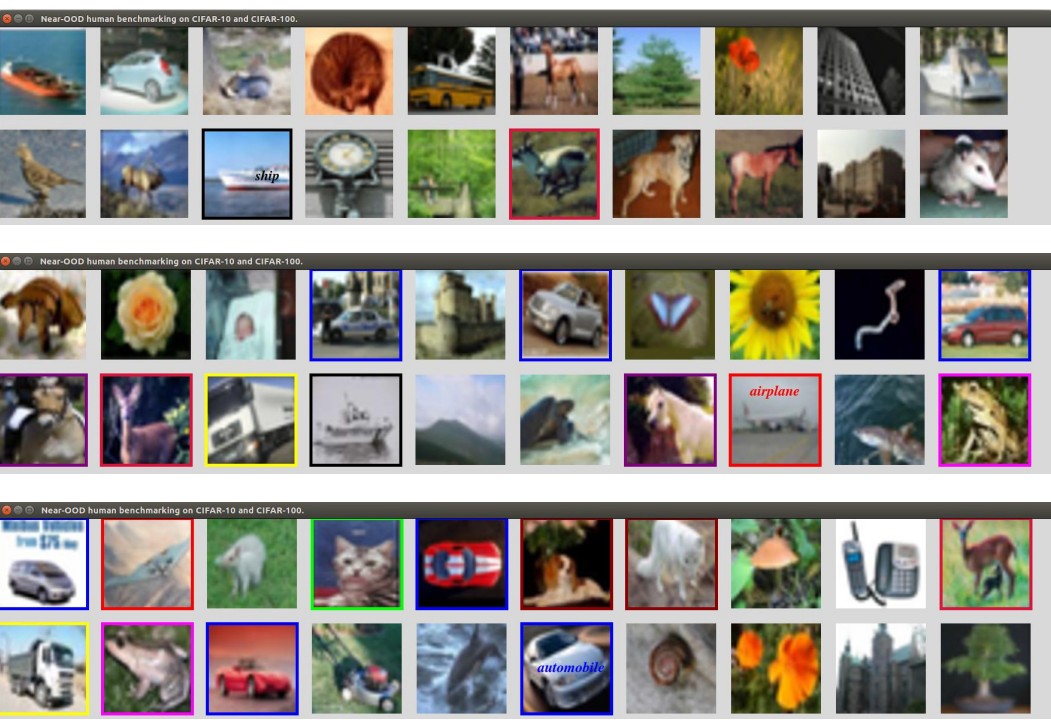

Figure 8: Graphical User Interface (GUI) for human benchmarking of near-OOD detection. The user clicks on images belonging to any of the 10 in-distribution CIFAR-10 classes. The user is shown 20 images at a time, and once they are done with their selection, they press a key and a new group of randomly chosen images (equal probability of CIFAR-10 and CIFAR-100) is shown. The figure shows 3 randomly chosen groups of 20 images, together with user-selected CIFAR-10 images framed by the color corresponding to their selected class.

In the case of CIFAR-10 and CIFAR-100 distributions (selecting the classes of CIFAR-10), the user clicks on all images belonging to each of the 10 classes: *airplane, automobile, bird, cat, deer, dog, frog, horse, ship, truck*. This is done without any exposure to the training set or examples of any of the classes, based only on the class names and using the fact that people are generally very familiar with the semantic concept of these labels. Coincidentally, this is quite similar to the kind of familiarity large pre-trained transformers gain by the breath of their pre-training.

We calculated the AUROC for the CIFAR-10 / CIFAR-100 task on their test sets, where choosing any of the 10 classes was acceptable as a valid CIFAR-10 selection by the user. The results are shown in Table 6. The average AUROC weighted by the number of images in each trial is AUROC $95.90\%$.

# B    Visualizing the effect of outlier exposure on fine-tuned pre-trained models

Figure 9 shows the outlier score on near-OOD task. Intuitively, the embeddings obtained by fine-tuning a pre-trained transformer are well-clustered, so just a handful of known outliers can significantly improve OOD detection.

Table 6: Author's results on CIFAR-100 vs CIFAR-10 distinguishing task.

| Date | Number of images | AUROC |
|------|------------------|-------|
| April 7 2021 | 1140 | 96.14% |
| April 25 2021 | 1700 | 95.67% |
| April 27 2021 | 940 | 96.03% |

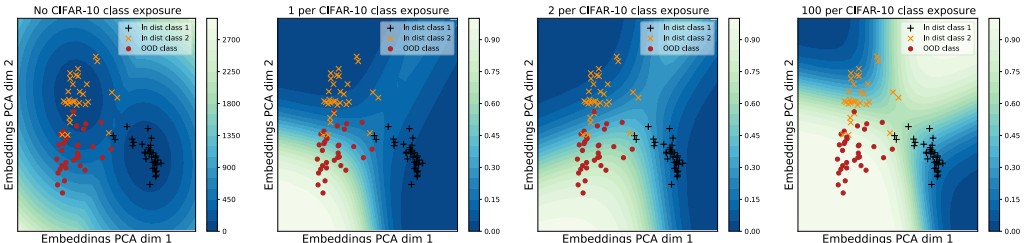

Figure 9: The effect of outlier exposure (CIFAR-10) on ViT a model fine-tuned on CIFAR-100. Each panel shows the same two-dimensional PCA cut of the space of embeddings. Examples of 2 in-distribution (CIFAR-100) classes (black plus signs = "bus", yellow crosses = "pickup truck"), and 1 out-of-distribution (CIFAR-10) class (red dot = "automobile"). The left-most plot shows Mahalanobis OOD score as the color coding. The more OOD outlier exposure (second to left to right), the better aligned the OOD probability contours with the underlying classes.

## C    Additional results for visual OOD detection

We tested our OOD detection on several more tasks in addition to CIFAR-$\{10, 100\}$ → CIFAR-$\{100, 10\}$ in Table 1. We used the *Describable Textures dataset* (DTD, or Textures in our tables) [Cimpoi et al., 2014], *Places365-Standard* (Places365 in our tables) [Zhou et al., 2017] and SVHN Netzer et al. [2011] datasets as provided by tensorflow datasets.[8] In all cases we use the test set of each dataset.

Besides the area under the receiver operating characteristic curve (AUROC), we also evaluate OOD performance using the area under the precision-recall curve (AUPRC) and the false positive rate at N% true positive rate (FPRN), as in Hendrycks et al. [2018]. Since our goal is to detect OOD, we treat OOD test set as the positive set and in-distribution test set as the negative set. AUROC and AUPRC are threshold independent, evaluating the overall OOD performance across multiple thresholds. AUROC is also sample size independent, while AUPRC is sensitive to detect imbalance between positive and negative sets. FPRN computes the false positive rate at which $N\%$ of OOD data is recalled. As a common practice, we set $N = 95$.

Table 7: ImageNet-21k pre-trained Vision Transformer fine-tuned on the in-distribution training set. More datasets in addition to Table 1.

| In-distribution | fine-tuned test accuracy | Out-distribution | Maha. AUROC ↑ | Maha. AUPRC↑ | Maha. FPR95↓ | MSP AUROC↑ | MSP AUPRC↑ | MSP FPR95↓ |
|-----------------|--------------------------|------------------|---------------|--------------|--------------|------------|------------|------------|
| CIFAR-100 | 91.71% | CIFAR-10 | 96.23% | 96.32% | 18.73% | 92.13% | 92.57% | 37.73% |
| | | Textures | 99.03% | 96.53% | 4.27% | 96.28% | 99.22% | 17.30% |
| | | SVHN | 97.80% | 98.87% | 8.42% | 97.15% | 93.61% | 13.05% |
| | | Places365 | 93.95% | 99.78% | 29.22% | 88.37% | 39.57% | 51.02% |
| CIFAR-10 | 98.70% | CIFAR-100 | 98.52% | 98.70% | 6.89% | 97.79% | 97.72% | 9.76% |
| | | Textures | 99.97% | 99.83% | 0.05% | 99.59% | 99.92% | 1.73% |
| | | SVHN | 99.58% | 99.82% | 1.90% | 98.77% | 97.75% | 4.03% |
| | | Places365 | 98.51% | 99.95% | 4.54% | 97.14% | 50.92% | 10.13% |

The additional results are shown in Table 7. SVHN and Textures are far-OOD tasks and our methods achieve AUROC of around $98\%$ or higher. Places365 is a harder OOD task, see the discussion in

---
[8]https://www.tensorflow.org/datasets

[Winkens et al., 2020] where they compute Confusion Log Probability (CLP) score for different OOD datasets and demonstrate that CIFAR-100 vs CIFAR-10 and CIFAR-* vs Places365 are more difficult OOD detection tasks than CIFAR-* vs SVHN and CIFAR-* vs Textures. On CIFAR-100 vs Places365, we achieve 93.9% (Winkens et al. [2020] report 82%) and on CIFAR-10 vs Places365, we achieve 98.5% (Winkens et al. [2020] report 95%).

To ensure that the approach works when training using high-resolution images, we also fine-tuned using ImageNet2012 and used Places365 as OOD. We are not aware of results using Mahalanobis distance for this dataset pair, but "Scaling Out-of-Distribution Detection for Real-World Settings" report 79% using MSP and 83% for MaxLogit (un-normalized logit) for ResNet-101. Using pre-trained ViT, we achieve 82.79% for MSP, 89.43% for MaxLogit, both of outperform the previously reported numbers. For completeness, we achieve 87.35% AUROC for Mahalanobis distance. See results in Table 8.

Table 8: ImageNet-21k pre-trained Vision Transformer fine-tuned on ImageNet-2012.

| In-distribution | fine-tuned test accuracy | Out-distribution | Maha. AUROC ↑ | MSP AUROC↑ | MaxLogit AUROC↑ |
|---|---|---|---|---|---|
| ImageNet-1k | 83.97% | Places365_small | 87.35% | 82.79% | 89.43% |

## C.1 Scaling of OOD performance with in-distribution test accuracy

In this section, we evaluate the relationship between fine-tuned in-distribution test accuracy and OOD detection performance. We looked at intermediate checkpoints during fine-tuning and observed the dependence of the in-distribution test accuracy and OOD detection performance (AUROC). We focused on the CIFAR-100 fine-tuning and the CIFAR-100 vs CIFAR-10 OOD detection task as it was the most challenging widely used benchmark we studied. The results are shown in Figure 10. We observe that the higher the in-distribution test accuracy, the higher the OOD detection AUROC. For the Vision Transformer, fine-tuning with and without augmentation leads to the same relationship between accuracy and OOD detection performance. However, we found that training with augmentation leads to a slower convergence. We found that the scaling relationship we identified on ViT-B_16 holds remarkably well even for large ViT models, such as ViT-L_16 as shown in Figure 10. As shown in Table 9, larger ViT models such as ViT-L_16 and ensemble of ViT models further improve OOD detection, improving the AUROC from 96.23% to approximately 98.11%, establishing a new state-of-the-art on this task.

Table 9: ImageNet-21k pre-trained ViT fine-tuned on the in-distribution training set.

| Model | In-distribution | fine-tuned test accuracy | Out-distribution | Mahalanobis AUROC | MSP AUROC |
|---|---|---|---|---|---|
| ViT-B_16 | CIFAR-100 | 90.95% | CIFAR-10 | 95.53% | 91.89% |
| R50+ViT-B_16 | CIFAR-100 | 91.71% | CIFAR-10 | **96.23%** | 92.08% |
| ViT-L_16 | CIFAR-100 | 94.73% | CIFAR-10 | **97.98%** | 94.28% |
| ViT ensemble | CIFAR-100 | — | CIFAR-10 | **98.11%** | 95.15% |

## C.2 Additional results for zero-shot OOD detection using CLIP

In Table 10 we extend the zero-shot OOD detection results for CLIP in Table 5 by using the names of the in-distribution classes and comparing them to the *the opposite of* concatenated with the same names. The performance is above random, but not significant compared to even using just the in-distribution names, as shown in Table 5.

## C.3 Qualitative analysis of OOD detection using ViT

In this section we present some qualitative failure cases of OOD detection. The results for the CIFAR-100 (in-distribution) vs CIFAR-10 (out-distribution) experiment are shown in Figure 11. For the CIFAR-10 images that are mistakenly classified as in-distribution (CIFAR-100), the top images

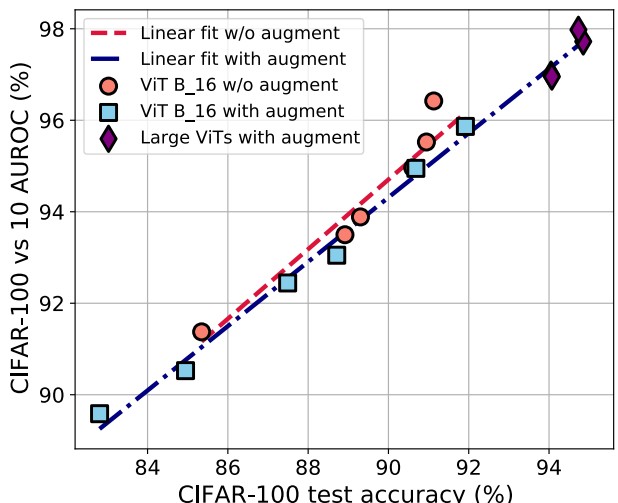

Figure 10: The dependence of OOD performance on in-distribution test accuracy for ViT models fine-tuned on CIFAR-100. Square markers represent different checkpoints during fine-tuning (lower accuracy corresponds to fewer steps of fine-tuning). The purple markers represent larger ViTs from Steiner et al. [2021]. The higher the accuracy on the in-distribution, the better the OOD performance. The blue linear fit uses only the square datapoints, but intersects the purple datapoints (large ViTs) well, suggesting a robust scaling.

Table 10: Zero-shot OOD detection using image-text multi-modal models. In Table 5 we show the performance of CLIP in the zero-shot OOD regime with candidate labels that are the names of the classes of the in-distribution, and optionally the names of the classes of the out-distribution dataset. In this table we present results using the names of the in-distribution classes as compared to *the opposite* concatenated with the same names.

| Distribution 1 | Distribution 2 | Labels 1 | Labels 2 | AUROC |
|---|---|---|---|---|
| CIFAR-100 | CIFAR-10 | CIFAR-100 names | "the opposite of" +CIFAR-100 names | 61.37% |
| CIFAR-10 | CIFAR-100 | CIFAR-10 names | "the opposite of" +CIFAR-10 names | 71.894% |

are actually mislabeled images from the CIFAR-10 test set (a fox labeled as a *cat*, and a kangaroo labeled as a *deer*). They are followed by *automobiles* and *trucks* (both CIFAR-10 classes) that are seen as *buses* and *streetcars* (similar CIFAR-100 classes). In those cases, the distinction could be unclear even to a human.

We show several qualitative failure cases for the CIFAR-100 (in-distribution) vs SVHN (out-distribution) experiment in Figure 12. The out-distribution images that are most like CIFAR-100 (Figure 12a) are numbers "2", "5" and "6" written in an especially wiggly font, sometimes with plastic-like filler, giving them a very worm-like quality that the model perceives. The last image we show gets perceived as a cloud, likely due to its deep blue background and low resolution. In Figure 12b we show the images from the in-distribution that have the closest embedding vector to the SVHN mistakes. They do look very similar.

Overall the mistakes our models make are semantically meaningful to a human. For CIFAR-10, they contain mislabeled examples of the CIFAR-10 test set that should not have been included in the first place, or genuine imperfect categories such as *automobile*, *truck*, *bus*, and *streetcar*, which are even harder to distinguish in the $32 \times 32$ resolution. For SVHN, the mistakes we get are also very plausible to a human.

Figure 13 and Figure 14 show examples of the most confusing pairs of (OOD class (CIFAR-10), in-distribution class (CIFAR-100)). We looked at the OOD examples with the lowest Mahalanobis distances, e.g. the ones that are the most in-distribution-like to the network at hand (ViT fine-tuned on the CIFAR-100 train set). Going from the smallest distance up, we sorted these images based on the

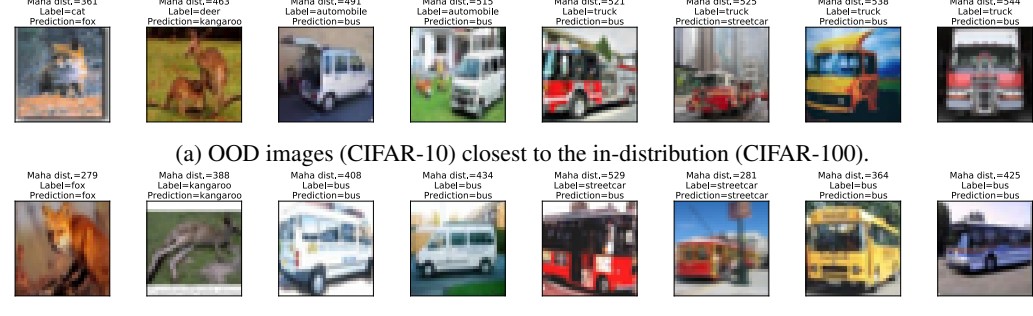

(a) OOD images (CIFAR-10) closest to the in-distribution (CIFAR-100).

(b) The in-distribution (CIFAR-100) images with the closest embedding vector to images in Figure 11a.

Figure 11: CIFAR-100 (in-distribution) vs CIFAR-10 (out-distribution): Images from the out-distribution that reached the lowest Mahalanobis distances from the in-distribution (Figure 11a) and the in-distribution images with the most similar embedding vectors (Figure 11b). The comparison to the closest in-distribution images demonstrates that OOD detection in these particular cases might be failing due to the limitations of the labelling in the original dataset and genuine semantic ambiguity of some classes. The first two images in Figure 11a do not belong to CIFAR-10 and have likely been included by accident.

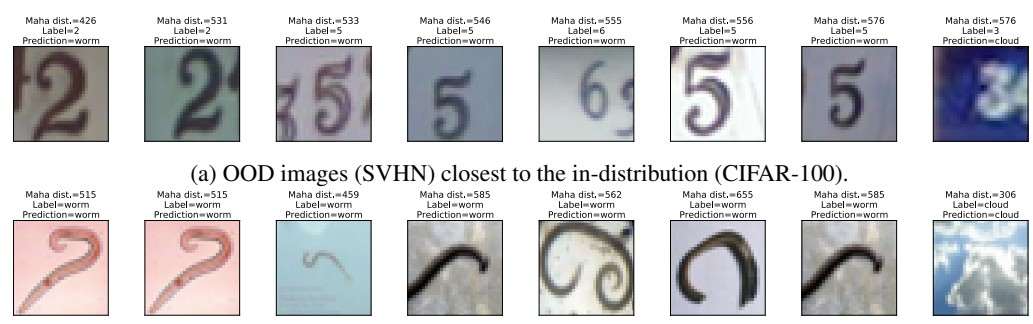

(a) OOD images (SVHN) closest to the in-distribution (CIFAR-100).

(b) The in-distribution (CIFAR-100) images with the closest embedding vector to images in Figure 12a.

Figure 12: CIFAR-100 (in-distribution) vs SVHN (out-distribution): Images from the out-distribution that reached the lowest Mahalanobis distances from the in-distribution (Figure 12a) and the in-distribution images with the most similar embedding vectors (Figure 12b). The most similar images demonstrate that the some of the digits genuinely look like worms.

tuple of which CIFAR-10 class the OOD image came from, and which class the CIFAR-100 test set image that has the most similar embedding vector to it belongs. We show examples that come from classes related to **vehicles** in Figure 13, and to **animals** in Figure 14. Some of the least-OOD-like CIFAR-10 examples shown are literal mistakes, and they should not have been included in CIFAR-10 at all (e.g. the first column "truck" being actually a tractor in Figure 13, or the first column "cat" in Figure 14 being actually a fox). Many of the classes have a genuine semantic overlap, especially among the vehicles. For example, the CIFAR-10 class "automobile" has a large number of vans, that are also included in the CIFAR-100 class "bus". The CIFAR-10 class "truck" and the CIFAR-100 class "pickup truck" share equally mutually similar vehicles.

# D    Additional results for genomics OOD detection

We report additional results for genomics in Table 11 and Table 12.

Table 11: Genomics OOD detection using pre-trained BERT and fine-tuned on the in-distribution training set. The error bars shown are standard deviations over 3 runs.

| Model | Test Accuracy | Mahalanobis | | | MSP | | |
|---|---|---|---|---|---|---|---|
| | | AUROC↑ | AUPRC↑ | FPR95↓ | AUROC↑ | AUPRC↑ | FPR95↓ |
| 1D CNN [Ren et al., 2019] | 85.93±0.11% | 64.75±0.73% | 60.25±0.82% | 77.76±0.84% | 65.84±0.46% | 62.24±0.31% | 89.79±0.18% |
| BERT pre-train and fine-tune | 89.84±0.00% | 77.49±0.04% | 78.79±0.06% | 68.22±0.13% | 73.53±0.03% | 73.86±0.03 | 85.39±0.07% |

Table 12: Few-shot outlier exposure for genomics OOD. The numbers here correspond to Figure 6. The error bar is the standard deviation of 3 runs.

| # OOD examples per class | w/o fine-tuning ignore outlier label | w/o fine-tuning use outlier label | w/ fine-tuning ignore outlier label | w/ fine-tuning use outlier label |
|---|---|---|---|---|
| 1 | 75.32±1.14% | 74.95±0.87% | 84.36±0.60% | 82.21±1.10% |
| 2 | 75.67±0.50% | 76.01±0.19% | 84.81±0.83% | 83.74±0.47% |
| 5 | 77.84±1.06% | 77.67±0.71% | 86.54±0.58% | 85.94±0.28% |
| 10 | 77.81±1.37% | 79.00±0.69% | 87.02±0.45% | 86.49±1.43% |
| 100 | 80.21±0.14% | 82.17±0.33% | 87.94±0.62% | 88.49±0.51% |

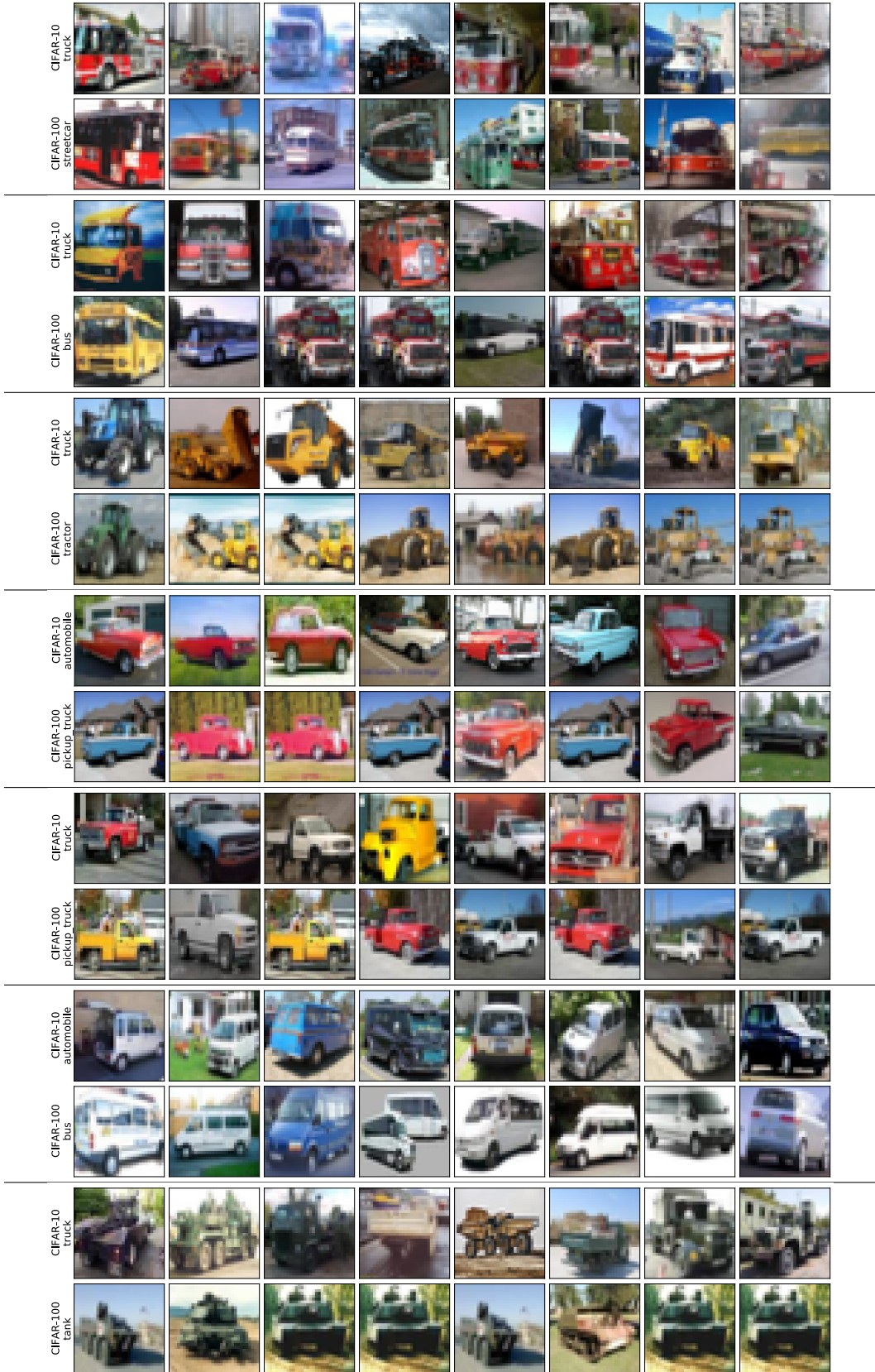

Figure 13: CIFAR-100 (in-distribution) vs CIFAR-10 (out-distribution), hard to distinguish pairs of classes involving **vehicles**: The plots show pairs of images, top row from the out-distribution (CIFAR-10) and the bottom row having the closest embedding vector to it from the in-distribution (CIFAR-100), for pairs of classes for which OOD images are the closest to the in-distribution (=hardest to distinguish). Our examples show that there is a genuine semantic overlap between some CIFAR-10 and CIFAR-100 classes that limits the OOD performance of any model.

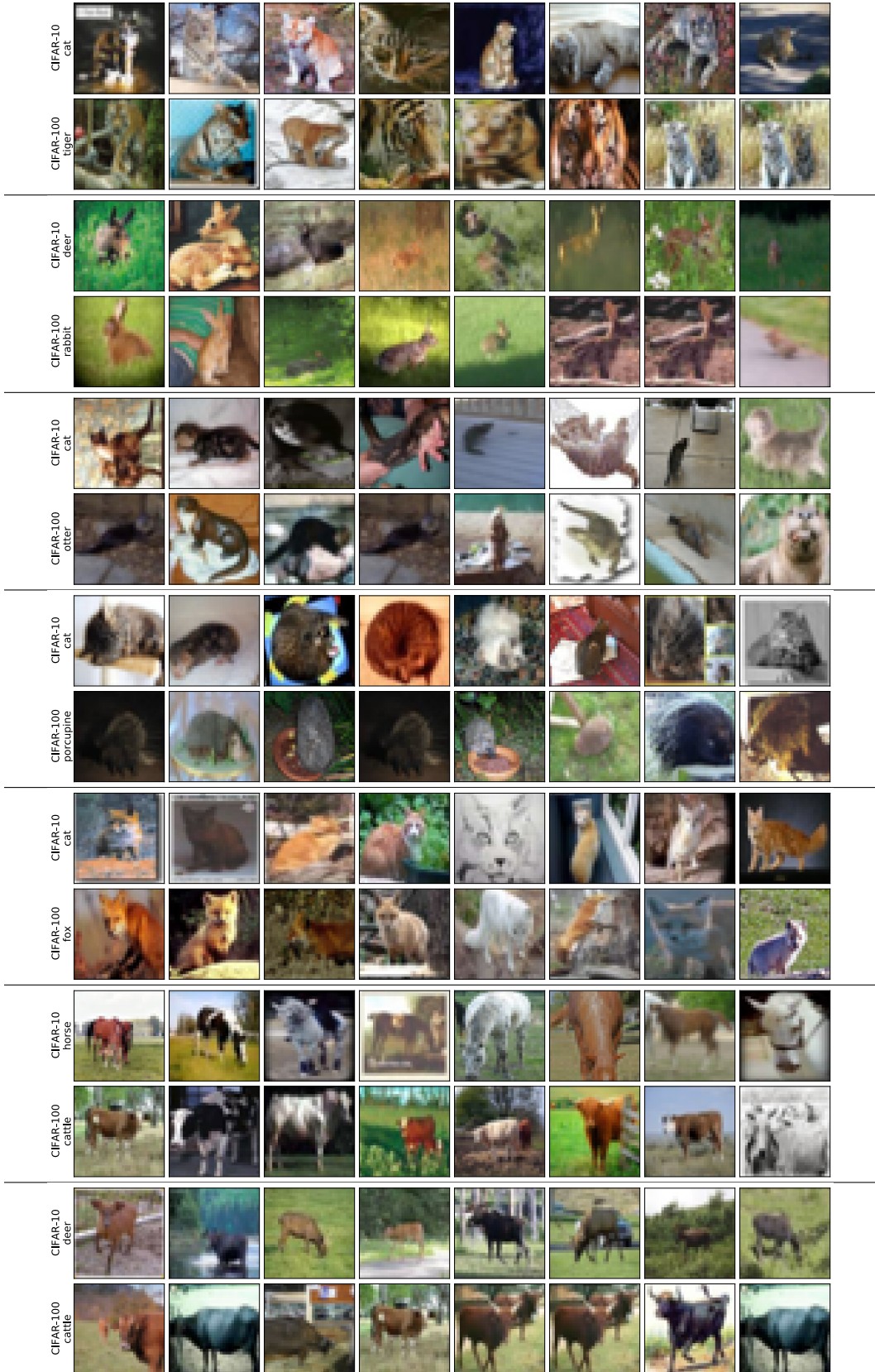

Figure 14: CIFAR-100 (in-distribution) vs CIFAR-10 (out-distribution), hard to distinguish pairs of classes involving **animals**: The plots show pairs of images, top row from the out-distribution (CIFAR-10) and the bottom row having the closest embedding vector to it from the in-distribution (CIFAR-100), for pairs of classes for which OOD images are the closest to the in-distribution (=hardest to distinguish). Our examples show that there is a genuine semantic overlap between some CIFAR-10 and CIFAR-100 classes that limits the OOD performance of any model.