# OpenReview forum: "Exploring the Limits of Out-of-Distribution Detection"
_NeurIPS.cc/2021/Conference — NeurIPS 2021 Poster_

### Official Review · Reviewer_nQCh · 2021-07-05

**Rating:** 7
**Confidence:** 4

**Summary:**

This paper finds and demonstrates that large-scale pre-trained transformers can significantly improve the state-of-the-art  on a range of near OOD tasks across different data modalities. The main contribution is that they find an efficient way to apply pre-trained transformer to OOD tasks and boost the performance significantly.

**Limitations And Societal Impact:**

Since the pretrained transformer is trained with large-scale dataset, while in this paper, the comparisons are primarily based on CIFAR10 and CIFAR100, which is too small to convince people thoroughly. It would be suggested that the authors could apply the same strategy to various OOD tasks and justify the efficiency and robustness of the proposed method.

**Main Review:**

Though not surprised about this finding and conclusion, it is impressive that the authors provide detailed study and experiments on different comparisons. The paper is easy to read and follow, this would provide an important method to deal with OOD based on pretrained transformers. Also, beyond OOD, this paper could also bring some insights about how powerful the pretrained transformer will be for various vision tasks.

Strength:
I like the analytical experiments in the paper, such as Figure 1. The figure is very clear and easy to understand.
The paper is well-written and easy to follow.
The proposed method is simple but effective and could bring some insights for future research.

Concerns:
In line 123, the authors mention that `Self-supervised learning techniques have also been shown to improve OOD robustness`, what’s the underlying difference between the self-supervised methods and pretrained model. Does large-scale data-driven pretrained model should be the main issue for OOD tasks? Is there some cases that pretrained strategy works worse than self-supervised learning techniques? If yes or no, could the authors explain it?

In line 134, the authors claim that `We found that we get better generalization and higher quality embeddings when we do not use data augmentation for fine-tuning.` I am curious about which data augmentation the authors used for the experiments, is there any explanation for this part?

In general the whole paper is very clear and easy to follow. While I still have concerns regarding only conduct OOD experiments on CIFAR10 and a few relevant very small datasets with very low resolution ($32 \times 32$). Since the pretrained transformer is trained with large-scale datasets, so it is possible that the transformers already learn some related patterns. The improvement is significant, but wondering have the authors tried on other dataset with higher resolutions. If yes, how it goes with the pre-trained model? I feel like it might be harder to achieve a much higher performance on those dataset based on the pretrained model using current straight-forward strategy. I will change my scores based on the explanation of this part.

------------------- post rebuttal-----------------------
Thanks for the rebuttal, I am satisfied with the explanation, but still conservative about the potential usage of this method for the community, therefore, I keep my score unchanged.

**Time Spent Reviewing:**

6-7

---

> ### Author Response · Authors · 2021-08-10
> **Response to Reviewer nQCh**
>
> Thank you for your review and positive feedback!
>
> **“In line 123, the authors mention that Self-supervised learning techniques have also been shown to improve OOD robustness, what’s the underlying difference between the self-supervised methods and pretrained model. Does large-scale data-driven pretrained model should be the main issue for OOD tasks? Is there some cases that pretrained strategy works worse than self-supervised learning techniques? If yes or no, could the authors explain it?”**
> Self-supervised learning is regarded as one of the pretraining approaches (note also that terminology is sometimes used interchangeably in the literature). If the label information is available during pre-training, we can use supervised pre-training, e.g. as in ViT model. If the label information is not available but the pair information between image and text is available, we can use the self-supervised pre-training. If none of them are available, we can generate the paired data for self-supervised pre-training as DINO [4].
>
> **“ Is there some cases that pretrained strategy works worse than self-supervised learning techniques?”**
> To compare different strategies, we evaluate the OOD performance for the self-supervised DINO ViT model [4]. Even though DINO ViT-B_16 is pre-trained without labels, the AUROC is significantly higher than the current SOTA for CIFAR-100 vs CIFAR-10. However, supervised pre-training further improves performance.
> [4] Caron, Mathilde, et al. "Emerging properties in self-supervised vision transformers." arXiv preprint arXiv:2104.14294 (2021).
>
> | Model                 | In-distribution | Fine-tuned test accuracy | Out-distribution | Mahalanobis AUROC | MSP AUROC |
> |-----------------------|-----------------|--------------------------|------------------|-------------------|-----------|
> | DINO ViT-B_16         | CIFAR-100       | 88.95%                   | CIFAR-10         | 88.78%            | 81.25%    |
> | ViT-B_16 (early stop) | CIFAR-100       | 88.71%                   | CIFAR-10         | 93.05%            | 88.82%    |
> | ViT-B_16              | CIFAR-100       | 90.95%                   | CIFAR-10         | 95.53%            | 91.89%    |
> | R50+ViT-B_16          | CIFAR-100       | 91.71%                   | CIFAR-10         | 96.23%            | 92.08%    |
>
>
> **“In line 134, the authors claim that We found that we get better generalization and higher quality embeddings when we do not use data augmentation for fine-tuning. I am curious about which data augmentation the authors used for the experiments, is there any explanation for this part?”**
>
> We use the default data augmentation used in ViT model, which involves random image flips and crop. Our understanding is that since pre-training has already learned robust features and the pre-training has used data augmentation methods, during fine-tuning it may not be necessary to have additional data augmentation.
>
> **Since the pretrained transformer is trained with large-scale datasets, so it is possible that the transformers already learn some related patterns. The improvement is significant, but wondering have the authors tried on other dataset with higher resolutions. If yes, how it goes with the pre-trained model? I feel like it might be harder to achieve a much higher performance on those dataset based on the pretrained model using current straight-forward strategy. I will change my scores based on the explanation of this part.”**
>
> The inputs to the ViTs used are 384x384x3, and we’re therefore always interpolating the input images (for example the 32x32x3 CIFAR-10/100) to the full resolution using a bicubic interpolation. Therefore the images that the model sees are always relatively high resolution, in the sense that they do not have straight, easy-to-detect edges, but their originally lower resolution only shows as a potentially blurriness.
>
> In addition, in Appendix C Table 6 with caption “ ImageNet-21k pre-trained Vision Transformer fine-tuned on the in-distribution training set. More datasets in addition to Table 1“, we present results for CIFAR- vs Places365, which are higher resolution by default. There, we finetune the model on either CIFAR-10 or CIFAR-100 (rescaled, low resolution originally), which, if you were right, should be least favourable mode for our model to work in -- “low” resolution finetuning (rescaled to 384x384) + high-resolution task -- yet the model performs really well.
>
> To ensure that the approach works when training using high-resolution images, we also finetuned using ImageNet2012 and used Places365 as OOD. We are not aware of results using Mahalanobis distance for this dataset pair, but “Scaling Out-of-Distribution Detection for Real-World Settings” by [Hendrycks et al. 2020] report 79% using MSP and 83% for MaxLogit (unnormalized logit baseline suggested in this paper by [Hendrycks et al. 2020]) for ResNet-101. Using pre-trained ViT, we achieve 82.79% for MSP, 89.43% for MaxLogit, both of which outperform the previously reported numbers. For completeness, we achieve  87.35% AUROC for Mahalanobis distance using ViT
>
> | Model    | In-distribution | Fine-tuned test accuracy | Out-distribution | Mahalanobis AUROC | MSP AUROC | MaxLogit AUROC |
> |----------|-----------------|--------------------------|------------------|-------------------|-----------|----------------|
> | ViT-B_16 | ImageNet2012    | 83.97%                   | Places365_small  | 87.35%            | 82.79%    | 89.43%         |
>
>
>
> **“primarily based on CIFAR10 and CIFAR100 ... It would be suggested that the authors could apply the same strategy to various OOD tasks and justify the efficiency and robustness of the proposed method.”**
> In addition to image modalities, we ran experiments on a different modality in Section 4, evaluating performance on OOD detection for genomics (motivated by a real-world application which requires high quality OOD detection for safe deployment). This is one of the hardest near-OOD benchmarks and we significantly push the performance there as well using similar strategy (pre-trained transformers + few-shot outlier exposure). Note that we focused on CIFAR10 and CIFAR100 for image modalities as this is a popular benchmark for near-OOD detection. As we showed above, our approach also works well for higher resolution image benchmarks and the methodology translates to difficult problems in different modalities.
>
>
> *We hope we have addressed most of your comments satisfactorily and kindly request you to consider increasing your score.*

---

> > ### Comment · Reviewer_nQCh · 2021-08-26
> > **Satisfied with the explanation**
> >
> > Thanks for the rebuttal, I am satisfied with the explanation, but still conservative about the potential usage of this method for the community, therefore, I keep my score unchanged.

---

### Official Review · Reviewer_qcJH · 2021-07-11

**Rating:** 6
**Confidence:** 5

**Summary:**

The paper experimentally examines the effect of using pre-trained (transformer) models for OOD detection and reports the following three results: i) pre-trained transformers lead to significant improvements on near-OOD benchmarks; ii) pre-trained transformers are well-suited for few-shot outlier exposure; iii) exploring pre-trained multi-modal image-text transformers in the zero-shot classification setting, just using the names of outlier classes as candidate text labels for CLIP can achieve good OOD detection performance.

**Limitations And Societal Impact:**

If my concerns above are not resolved, they will be limitations of this study.

**Main Review:**

I think this is a good paper that contains a lot of useful experimental results. But, on the other hand, I feel that its academic impact is slightly insufficient for a NeurIPS paper.

The three results can be summarized by the effectiveness of pretrained Transformer models for OOD detection. However, the effectiveness of pretraining for OOD detection is already known, as shown in, e.g., the following paper. (It is a problem that it is not cited.)

Hendrycks et al., Using Pre-Training Can Improve Model Robustness and Uncertainty, ICML2019

The paper’s claim (i) is that the two, i.e., pretraining + transformer, are important for improved performance. However, is it true? I suspect that not the transformer model but pretraining (on ImageNet21k) is the essence. Thus, even ResNet/EfficeinetNet models would show similarly good performance if they are pretrained on ImageNet21k and achieve similar ID classification accuracy. Table 1 shows comparisons with BiT-M R50x1, but they may not be fair since the compared models differ in the ID classification accuracy. Thus, the paper lacks sufficient evidence for the above claim. If my guess is true (i.e., the transformer is not necessary), then the results reported in this paper may not be so novel.

Regarding claim (ii), the method of few-shot outlier exposure is already known. It is unclear how novel it is to use transformer models for the backbone network.

Regarding claim (iii), the proposed method may be regarded as a new CLIP application but remains one of many of its applications.

In addition, one may question if we can still call it OOD detection even when we use pretrained models, especially those trained on a dataset sharing its classes with the OOD dataset. For instance, I suspect that most (or even all?) of CIFAR100 classes are the ImageNet21k classes. Can we call it OOD if an input belongs to a class learned in the pretraining?

**Time Spent Reviewing:**

4

---

> ### Author Response · Authors · 2021-08-10
> **Response to Reviewer qcJH**
>
> Thank you for your review.
>
> **“The three results can be summarized by the effectiveness of pretrained Transformer models for OOD detection. However, the effectiveness of pretraining for OOD detection is already known, as shown in, e.g., the following paper. (It is a problem that it is not cited.)”**
> Thank you for pointing out the above paper. We will definitely add a citation to “[Hendrycks et al 2019] Using Pre-Training Can Improve Model Robustness and Uncertainty”. Apologies for missing this. We cited a different paper which is also [Hendrycks et al., 2019] with a very similar title “Using self-supervised learning can improve model robustness and uncertainty”, but will also include this in the revision and add a discussion.
>
> There are several distinctions between our work and the above paper:
> 1. The pretraining in our work uses a large-scaled dataset compared to the previous work. Previous work uses Downsampled ImageNet (Chrabaszcz et al., 2017), which is the 1,000-class ImageNet dataset (Deng et al., 2009) resized to 32 × 32 resolution. We used ImageNet21k, which has 21k classes.
> 2. We focus more on the challenging near-OOD detection tasks and our goal is to explore the limit of this challenging task, using whatever techniques we can, and compare with human performance. The previous work evaluates on far OOD tasks such as CIFAR vs Gaussian noise, textures, and Places365 scene images.
> 3. We studied how the popular large vision transformer model’s performance on near-OOD problems, while the previous work uses 40-2 Wide Residual Networks.
>
>
> **“The paper’s claim (i) is that the two, i.e., pretraining + transformer, are important for improved performance. However, is it true? I suspect that not the transformer model but pretraining (on ImageNet21k) is the essence. Thus, even ResNet/EfficeinetNet models would show similarly good performance if they are pretrained on ImageNet21k and achieve similar ID classification accuracy. Table 1 shows comparisons with BiT-M R50x1, but they may not be fair since the compared models differ in the ID classification accuracy. Thus, the paper lacks sufficient evidence for the above claim. If my guess is true (i.e., the transformer is not necessary), then the results reported in this paper may not be so novel.”**
>
> Good point! We agree with you that to have a fair comparison between ResNet and ViT, both models need to have similar ID classification accuracy. So we evaluate the BiT-M R101x3 model which has 91.55%, a similar ID accuracy with 91.71% for R50+ViT-B_16 model. BiT-M model has a much lower AUROC 83.69% for MSP (ViT 92.08%), and a slightly low AUROC 90.10% for Mahalanobis method (ViT 96.23%). We observe that even when BiT and ViT have similar accuracy, ViT achieves higher AUROC for OOD detection. As an additional ablation, we also consider pre-training for MLP-mixer (another non-convolutional architecture), and find that they outperform BiT with comparable accuracy on OOD detection.
>
> | Model          | In-distribution | Fine-tuned test accuracy | Out-distribution | Mahalanobis AUROC | MSP AUROC |
> |----------------|-----------------|--------------------------|------------------|-------------------|-----------|
> | BiT-M R50x1    | CIFAR-100       | 87.01%                   | CIFAR-10         | 81.71%            | 81.15%    |
> | BiT-M R101x3   | CIFAR-100       | 91.55%                   | CIFAR-10         | 90.10%            | 83.69%    |
> | ViT-B_16       | CIFAR-100       | 90.95%                   | CIFAR-10         | 95.53%            | 91.89%    |
> | ViT-L_16       | CIFAR-100       | 94.73%                   | CIFAR-10         | 97.98%            | 94.28%    |
> | R50+ViT-B_16   | CIFAR-100       | 91.71%                   | CIFAR-10         | 96.23%            | 92.08%    |
> | MLP-Mixer-B_16 | CIFAR-100       | 90.40%                   | CIFAR-10         | 95.31%            | 90.22%    |
> | BiT-M R50x1    | CIFAR-10        | 97.47%                   | CIFAR-100        | 95.52%            | 85.87%    |
> | BiT-M R101x3   | CIFAR-10        | 97.36%                   | CIFAR-100        | 94.55%            | 85.34%    |
> | ViT-B_16       | CIFAR-10        | 98.10%                   | CIFAR-100        | 98.42%            | 97.68%    |
> | ViT-L_16       | CIFAR-10        | 99.16%                   | CIFAR-100        | 99.15%            | 98.49%    |
> | R50+ViT-B_16   | CIFAR-10        | 98.70%                   | CIFAR-100        | 98.52%            | 97.75%    |
> | MLP-Mixer-B_16 | CIFAR-10        | 97.58%                   | CIFAR-100        | 97.85%            | 96.28%    |
>
>
> **“ the method of few-shot outlier exposure is already known”**
> Could you please point us to a reference that explored few-shot outlier exposure?
> Outlier exposure was previously proposed, but to the best of our knowledge, we are not aware of any work that explores a few-shot outlier exposure similar to our setup. We believe we are the first to evaluate how OOD detection performance improves as a function of the number of known outliers.
> Previous work uses Tiny ImageNet (100,000 images and the test set has 10,000 images) as the dataset for outlier exposure, while our work focuses on the few-shot setting where we show only using 10 images per class we can achieve 99.46% AUROC. This setting is motivated by real-world applications which require high-quality OOD detection and teams are willing to collect a handful of known outlier examples (rather than just rely on modeling approaches) to improve safety.
>
> **“Regarding claim (iii), the proposed method may be regarded as a new CLIP application but remains one of many of its applications.”**
> Our claim was that multi-modal image-text pre-trained transformers offer the possibility of a weaker form of outlier exposure (just using the names of candidate labels without access to OOD images). To the best of our knowledge, such techniques have not been explored before and we believe this insight could be useful to the wider community.
>
> **“In addition, one may question if we can still call it OOD detection even when we use pretrained models, especially those trained on a dataset sharing its classes with the OOD dataset. For instance, I suspect that most (or even all?) of CIFAR100 classes are the ImageNet21k classes. Can we call it OOD if an input belongs to a class learned in the pretraining?”**
>
> Good point. It’s OOD from the point of view of the fine-tuned task. We will clarify this.
>
> In the genomics experiment, we actually pre-trained the transformer model in an unsupervised fashion using unlabeled in-distribution data. There is no overlapping between the pre-training data and the OOD test data. We see that the pre-training and fine-tuning approach significantly improved the OOD detection performance in this setting.
>
> For image experiment, we ran an additional ablation using an unsupervised pretrained model, as DINO [4]. Even though DINO ViT-B_16 is pre-trained without labels, the AUROC is significantly higher than the current SOTA for CIFAR-100 vs CIFAR-10 (~85%). Vision transformers are typically pre-trained in a supervised fashion and it’s unclear whether DINO is necessarily the optimal unsupervised pre-training baseline for this architecture; as the community develops better strategies for unsupervised pre-training and fine-tuning transformers, we believe that one can further improve OOD detection performance in this setting.
>
> | Model                 | In-distribution | Fine-tuned test accuracy | Out-distribution | Mahalanobis AUROC | MSP AUROC |
> |-----------------------|-----------------|--------------------------|------------------|-------------------|-----------|
> | DINO ViT-B_16         | CIFAR-100       | 88.95%                   | CIFAR-10         | 88.78%            | 81.25%    |
> | ViT-B_16 (early stop) | CIFAR-100       | 88.71%                   | CIFAR-10         | 93.05%            | 88.82%    |
> | ViT-B_16              | CIFAR-100       | 90.95%                   | CIFAR-10         | 95.53%            | 91.89%    |
> | R50+ViT-B_16          | CIFAR-100       | 91.71%                   | CIFAR-10         | 96.23%            | 92.08%    |
>
> [4] Caron, Mathilde, et al. "Emerging properties in self-supervised vision transformers." arXiv preprint arXiv:2104.14294 (2021).
>
> As an additional test, we evaluated performance of just pre-trained transformers (without any fine-tuning) by computing Mahalanobis distance. If the Imagenet-21k class hierarchy already separates CIFAR-100 vs CIFAR-100, this baseline should perform very well. However, ViT-B_16 achieves only 67.19% on CIFAR-100 vs CIFAR-10 (compared to more than 95% after fine-tuning on CIFAR-100) and only 84.88% on CIFAR-10 vs CIFAR-100 (compared to more than 98% after fine-tuning on CIFAR-10).
>
>
> **‘I think this is a good paper that contains a lot of useful experimental results. But, on the other hand, I feel that its academic impact is slightly insufficient for a NeurIPS paper.”**
>
> We were indeed shooting for a good paper that’s “useful” :) We believe that we have pushed the SOTA significantly on difficult “near-OOD” benchmarks and evaluated performance on two different modalities. We have run lots of interesting ablations which should be interesting to the community and also provide practitioners with a valuable guide to how to improve OOD detection performance on their problem. If we use citations as a measure of “academic impact”, the most well-cited NeurIPS papers are ones which push the SOTA and are simple to build on, not necessarily papers which introduce novelty for the sake of novelty.
>
>
>
> *We hope we have addressed most of your comments satisfactorily and kindly request you to consider increasing your score.*

---

> > ### Comment · Reviewer_qcJH · 2021-08-23
> > **Many useful experimental results**
> >
> > I appreciate the authors response and additional experiments. They mostly resolved my concern. I raised my initial rating.

---

### Official Review · Reviewer_dPMr · 2021-07-15

**Rating:** 8
**Confidence:** 4

**Summary:**

* The paper presents detailed experiments which show that pre-trained large scale transformers significantly improves OOD detection performance.
* The paper especially focuses on the case where in-distribution (InD) and OOD samples are close to each other (near OOD) (e.g. when CIFAR100 is InD and CIFAR10 is OOD, SoTA AUROC is improved from 85% to 96% ).
* It is demonstrated that transformers can yield better well-separated representations for each class on InD dataset compared to ResNet without even fine tuning on InD samples. Separation of the representations become even better after fine-tuning in InD dataset (see Fig. 1). This motivates that transformers can be better for OOD detection compared to ResNet-like architectures and improves the SoTA.
* The paper presents experiments in different settings on multiple datasets:
* First, OOD detection performance between CIFAR10 <-> CIFAR100 is evaluated. The results demonstrate that fine-tuning transformers significantly improves SoTA OOD detection performance. Additional experiments are presented where it is assumed that a few examples from each class of target OOD dataset is available (called outlier exposure). This leads to further boost in OOD detection performance.
* The second experiments are performed on a genomics dataset where InD dataset consist of 10 known bacteria classes and OOD dataset consist of another 60 bacteria classes which are not seen during training. The results are consistent with the CIFAR10-CIFAR100 experiments.
* Finally, experiments on multimodal text-image models are presented where candidate labels are used for OOD detection. The results show that it leads to significant performance improvement.

**Limitations And Societal Impact:**

- Please see the weaknesses section for some limitations of the paper.
- I don't see any potential negative social impact.


**Main Review:**

**Strengths**
- The paper presents extensive performance evaluation existing OOD detection methods on transformers architectures. The results presented in the paper demonstrate significant improvement over SoTA in challenging near OOD detection tasks.
- Despite of the fact that the paper doesn't propose a novel algorithm, the presented experiments are quite extensive.
- Transformers are popular architectures which achieve SoTA performance in variety of tasks. The paper demonstrate that they not only achieve high prediction performance, but also achieves significant OOD detection performance which makes it an ideal choice for safety-critical applications.

**Weaknesses**
I don't see any major weakness of the paper. There are just a few points that needs a bit clarification for me and a few suggestions for improvements.
- According to Fig. 1, ViT achieves perfect separation of classes without even fine-tuned on InD dataset. This is quite surprising to me because this mean that transformers can identify images from same classes without seeing any training examples. Is it because ViT is pre-trained on very large datasets? I think some intuition about how transformers can achieve this would be quite useful.
- In the outlier exposure experiments, a few examples from each class of target OOD are used to improve the OOD detection performance. However, target OOD dataset is usually unknown in real applications. So, although teams are willing to annotate some, they may not overlap with the target OOD dataset. For example, let's consider the scenario where CIFAR10 is InD, sample OOD data available for outlier exposure is CIFAR100, and OOD samples given to the model in test time is SVHN. Do the model assign SVHN samples to one of the InD or OOD classes?
- One generic way of performing outlier exposure could be using adversarial samples as OOD and testing how they generalise to various unseen OOD datasets.
- In lines 278-279, the paper mentions that the best model is checkpointed based on best validation AUROC. However, the best model is usually checkpointed based on the best prediction accuracy rather than the best OOD detection performance. How would the results change if the models were saved based on the best prediction accuracy? Do this affect the OOD detection performance significantly?

**Time Spent Reviewing:**

3

---

> ### Author Response · Authors · 2021-08-10
> **Response to Reviewer dPMr**
>
> Thank you for your detailed review and positive comments!
>
> **“According to Fig. 1, ViT achieves perfect separation of classes without even fine-tuned on InD dataset. This is quite surprising to me because this mean that transformers can identify images from same classes without seeing any training examples. Is it because ViT is pre-trained on very large datasets? I think some intuition about how transformers can achieve this would be quite useful.”**
> Yes, we believe that the representations learned by pre-trained transformers are more robust and more effective at preventing shortcut learning than NNs trained on just the dataset (without pre-training). We found that fine-tuning pre-trained models leads to tighter clusters (Figure 1 right) and improves OOD detection further.
>
>
> **“In the outlier exposure experiments, a few examples from each class of target OOD are used to improve the OOD detection performance. However, target OOD dataset is usually unknown in real applications. So, although teams are willing to annotate some, they may not overlap with the target OOD dataset. For example, let's consider the scenario where CIFAR10 is InD, sample OOD data available for outlier exposure is CIFAR100, and OOD samples given to the model in test time is SVHN. Do the model assign SVHN samples to one of the InD or OOD classes?”**
>
> Good question! We’ll answer the specific question first and address a more general question.
> On the specific CIFAR vs SVHN task: note that this is a far-OOD task and the models are really good already, so headroom for improvement is a bit limited. See results in Table 6 in Appendix C where we achieve AUROC of 99.58% on CIFAR-10 vs SVHN and 97.8% on CIFAR-100 vs SVHN. Furthermore, we looked at the “failures” of ViT in Figure 11 and found that SVHN images that are classified as in-distribution look like “worm” class in CIFAR-100.
> To answer a more general question of how choice of dataset for outlier exposure translates performance to other datasets, we ran the following experiments.
>
> **“How does few-shot outlier exposure work on unseen OOD test inputs?”**
>
> *Genomics experiment*: since the genomics benchmark dataset has validation OOD classes non-overlapping with the test OOD classes, we test the few-shot outlier exposure method on the setting where validation OODs are used as the source of few-shot outlier exposure during training, and evaluate on the test OODs. The results are shown in the following table. As a comparison, we also put the results reported in the paper here where test OOD classes overlapped with the training few-shot OOD classes. It can be seen that when OOD classes for few-show exposure are different from the few-shot OOD, the performance decreases a bit, but the trend is the same and they all perform better than the case without OOD exposure.
>
> |                                                                 |                                | Outlier examples per OOD class |        |        |        |        |
> |-----------------------------------------------------------------|--------------------------------|:------------------------------:|:------:|:------:|:------:|:------:|
> |                                                                 |                                | 1                              | 2      | 5      | 10     | 100    |
> | Test OOD classes != training OOD classes                        | Finetuned, use outlier label   | 80.80%                         | 79.91% | 81.70% | 82.12% | 83.43% |
> |                                                                 | Finetuned ignore outlier label | 79.72%                         | 82.61% | 82.98% | 82.33% | 84.06% |
> | Test OOD classes = training OOD classes (reported in the paper) | Finetuned, use outlier label   | 85.19%                         | 84.91% | 86.36% | 86.42% | 88.56% |
> |                                                                 | Finetuned ignore outlier label | 83.08%                         | 84.34% | 85.93% | 87.19% | 87.87% |
>
>
> *Image experiments* For images, we took R50+ViT_B-16 finetuned on CIFAR-10 achieving test accuracy=98.73%
> We used CIFAR-100 as the OOD dataset, and used only the classes 0-79 for outlier exposure, and evaluated the performance AUROC on CIFAR-10 vs CIFAR-100 (classes 80-99 only). Even though the model is exposed to some classes, it improves OOD-detection performance on other classes.
>
> |                                                                 | Outlier examples per OOD class |               |
> |-----------------------------------------------------------------|:------------------------------:|:-------------:|
> |                                                                 | 1                              | 2             |
> | Test OOD classes != training OOD classes                        | 98.09%                         | 98.71%        |
> | Test OOD classes = training OOD classes (reported in the paper) | 98.96 ± 0.05%                  | 99.11 ± 0.04% |
>
>
> **“One generic way of performing outlier exposure could be using adversarial samples as OOD and testing how they generalise to various unseen OOD datasets.”**
>
> Thanks for the suggestion, we believe these are complementary to our proposed ideas. We will explore this in future work.
>
> **"In lines 278-279, the paper mentions that the best model is checkpointed based on best validation AUROC. However, the best model is usually checkpointed based on the best prediction accuracy rather than the best OOD detection performance. How would the results change if the models were saved based on the best prediction accuracy? Do this affect the OOD detection performance significantly?"**
> Thank you for your comment. We noticed that in particular in the genomics experiment, the OOD performance for Mahalanobis distance changes during training and the trend may not concur with the prediction accuracy/ loss.  We report the performance when the model checkpoint is selected based on lowest validation loss or highest validation accuracy below. In all the cases, the BERT pretrain and finetune model has better performance than the baseline.
>
> Model checkpoint selected based on the lowest validation loss
>
> |                              | Test accuracy | Mahalanobis AUROC | MSP AUROC |
> |------------------------------|---------------|-------------------|-----------|
> | 1D CNN                       | 0.8433        | 0.5748            | 0.6491    |
> | BERT pre-train and fine-tune | 0.8715        | 0.6617            | 0.6969    |
>
> Model checkpoint selected based on the highest validation acc
>
> |                              | Test accuracy | Mahalanobis AUROC | MSP AUROC |
> |------------------------------|---------------|-------------------|-----------|
> | 1D CNN                       | 0.8587        | 0.5291            | 0.6614    |
> | BERT pre-train and fine-tune | 0.8951        | 0.5625            | 0.7162    |
>
>
> *We hope we have addressed most of your comments satisfactorily and kindly request you to consider increasing your score.*

---

> > ### Comment · Reviewer_dPMr · 2021-08-22
> > **I support this paper for publication**
> >
> > I would like to thank authors for the rebuttal and addressing my comments. The paper presents quite interesting results with very detailed analysis. So, I retain my positive view for this paper and support its acceptance for Neurips2021.

---

### Official Review · Reviewer_s3xZ · 2021-07-22

**Rating:** 6
**Confidence:** 5

**Summary:**

This paper studies the few-shot outlier detection setting where a few examples from outlier classes are provided to the ODD detector.  The main motivation behind this work are situations that require high level of ODD detection. It is shown that classifiers trained by fine-tuning large-scale pre-trained transformers are significantly better at near-ODD detection. Simulations are performed on well-known data sets for visual classification and genomic sequences. Performance evaluation are reported in term of AUROC scores. However, it would have been more convenient to report results in terms of FPR at 95% scores which allow to better understand if the performance gains with respect to similar methods are relevant. The reported gains are not very significant but consistent.


**Ethical Concerns:**

There is no ethical concerns to be reported.

**Limitations And Societal Impact:**

I do not think the authors addressed the main limitation related to the impossibility of disposing of the ODD samples corresponding to the testing distribution. As for the societal impact, the investigated topic (e.g., ODD detection) is very relevant and can have significant impact on real-word systems.

**Main Review:**

The main limitation of this work relies on the assumption that out-of-distribution samples can be labelled and be available to train the detector. I strongly believe this is not the case in many of practical scenarios where these samples evolve over time and this cannot be given in advance to the learner. On the other hand, the paper lacks of novelty in terms of exceptions for a NeurIPS work. The proposed method relies on the addition of the soft-probabilities of the concatenated  predictions for in and out distributions. Finally, it is not clear how much this method improve on state-of-the-art methods since most of the existent literature on ODD detection does not assume of the presence of ODD samples from the same distribution than for testing. This also brings the question of how much the present method degrades  if the available ODD samples do not correspond to the same testing distributions. Additional simulations would have been beneficial to clarify this issue. Finally, I believe that FPR at 95% scores are much more relevant than  AUROC scores which have not been reported.

**Time Spent Reviewing:**

4 hours

---

> ### Author Response · Authors · 2021-08-10
> **Response to Reviewer s3xZ**
>
> Thank you for the review.
>
> **“Summary: This paper studies the few-shot outlier detection setting where a few examples from outlier classes are provided to the ODD detector.”**
>
> We’d like to clarify that we have significant additional contributions than just the few-shot setting pointed out in this summary. We have two other sets of experiments, please see lines 64-74 in the text for a summary of our contributions.
> 1. We explore near-OOD in Section 3.1 and Section 4 on two different data modalities. Note that this does not assume access to any outlier data, and we significantly improve over existing SOTA even in this setting. “We show that pre-trained transformers lead to significant improvements on near-OOD benchmarks. Concretely, we improve the AUROC of OOD detection on CIFAR-100 → CIFAR-10 from 85% (current SOTA) to more than 96% using ViT pre-trained on ImageNet-21k, and improve the AUROC on a genomics OOD detection benchmark from 66% (current SOTA) to 77% using BERT”.
> 2. Furthermore, we explore multi-modal models in Section 5. “We explore OOD detection for pre-trained multi-modal image-text transformers in the zero-shot classification setting, and show that just using the names of outlier classes as candidate text labels for CLIP can achieve AUROC of 94.8% on CIFAR-100 vs CIFAR-10 task. On easier far-OOD tasks such as CIFAR-{100, 10} → SVHN, we achieve AUROC of 99.6% and 99.9% respectively​​”.
>
>
>
> **“The main limitation of this work relies on the assumption that out-of-distribution samples can be labelled and be available to train the detector. “**
> As mentioned above, the assumption is only used in the few-shot setting.
> We have two other sets of experiments which do not assume access to out-of-distribution samples. Section 5 experiments highlight that multi-modal models can leverage weaker forms of outlier exposure.
>
>
> **“Finally, it is not clear how much this method improve on state-of-the-art methods since most of the existent literature on ODD detection does not assume of the presence of ODD samples from the same distribution than for testing.”**
> For the scenario where the OOD samples are not available, we show that our pre-trained and finetuned ViT model improves the AUROC to 96%, compared with the best of the previous baseline 85% (Table 1, Figure 2). See also our experiments in genomics in Section 4.
>
> **“I believe that FPR at 95% scores are much more relevant than AUROC scores which have not been reported”**
> We present the FPR95 numbers (and AUCPR) in Appendix C Table 6 for ViT models and Appendix D Table 7 for the genomics model. We had to move them to the Appendix due to space constraints. The ordering of methods is consistent irrespective of whether we use AUROC (higher is better) or FPR95 (lower is better).
> For the scenario where OOD dataset is not used for training, our ViT model achieves 18.73% FPR95 for CIFAR-100 vs CIFAR-10 task, and 6.89% for CIFAR-10 vs CIFAR-100 task. We did not find the FPR95 score reported in the hybrid model paper [1] which has the highest AUROC among the previous baselines. For the MSP baseline, it is reported to have 64.9% and 43.5% for CIFAR-100 vs CIFAR-10 and CIFAR-10 vs CIFAR-100 tasks respectively (Table 7 in Appendix A in [2]). We see significant reduction in FPR95 as well.
>
> [1] Zhang, Hongjie, et al. "Hybrid models for open set recognition." European Conference on Computer Vision. Springer, Cham, 2020.
>
> [2] Hendrycks, Dan, Mantas Mazeika, and Thomas Dietterich. "Deep anomaly detection with outlier exposure." arXiv preprint arXiv:1812.04606 (2018).
>
> **“The reported gains are not very significant”**
> We respectfully disagree. Near-OOD detection is one of the hardest OOD detection tasks for NNs. We show ~10% improvement in AUROC on two different modalities (AUROC for CIFAR-100 vs CIFAR-10 improved from 85% (current SOTA) to more than 96% using ViT, and AUROC for a genomics OOD detection benchmark improved from 66% (current SOTA) to 77%”. With few-shot outlier exposure, we further improve performance to 99% and 86% improvement. We believe that these are significant gains over the current SOTA.
>
> See also comments by other reviewers on the significance of our results. Reviewer dPMr “The paper demonstrate that they not only achieve high prediction performance, but also achieves significant OOD detection performance “ and Reviewer nQCh  “The main contribution is that they find an efficient way to apply pre-trained transformer to OOD tasks and boost the performance significantly.”
>
> **“I strongly believe this is not the case in many of practical scenarios where these samples evolve over time and this cannot be given in advance to the learner.”**
> First, we would like to reiterate that we have multiple contributions in the paper that do not assume access to OOD data (Section 3.1 and Section 4). We also explore weaker forms of outlier exposure for CLIP (by leveraging just names of outlier classes in Section 5 that relax this assumption.
> To address this comment, the few-shot outlier exposure setting was actually motivated by our collaborations with real world applications. From our experience, we do have scenarios where *a handful of OOD samples are available*, which is precisely why we evaluated improvements as a function of the number of known outliers.
> 1. For real-world applications which require high-quality OOD detection for safe deployment, teams are willing to collect a handful of known outlier examples (rather than just rely on modeling approaches) to improve safety.
> 2. Another setting is the case where models are being continuously re-trained; once an outlier is detected, it is desirable to include that in the training corpus quickly to encourage the model to correctly detect similar examples as outliers in the future. So we explore the few-shot outlier exposure set up, and we show that we can improve the AUROC to 99.46% using only 10 samples per class.
> Whether this assumption is practical is a subjective assessment, and based on our individual experiences. We wanted to push the limits of performance and demonstrate how much improvement can be obtained by few-shot outlier exposure. We believe this will be a useful addition to the toolkit. Practitioners can see the results and decide for themselves.
>
> **“This also brings the question of how much the present method degrades if the available ODD samples do not correspond to the same testing distributions. Additional simulations would have been beneficial to clarify this issue. “**
> We ran some additional tests to test this and found that few-shot outlier exposure improves OOD detection on unseen classes as well. Please see “How does few-shot outlier exposure work on unseen OOD test inputs?” in response to reviewer dPMr  below.
>
>
> *We hope we have addressed most of your comments satisfactorily and kindly request you to consider increasing your score.*

---

> > ### Comment · Reviewer_s3xZ · 2021-08-23
> > **Reply from authors about my concerns**
> >
> > I would like to thank to authors for the detailed rebuttal. I read the reviews of other reviewers and the rebuttal. I think authors addressed my of the concerns raised by the reviewers. I updated my score.

---

### Decision · Program_Chairs · 2021-09-27

**Decision:**

Accept (Poster)

**Comment:**

The reviewers are in agreement that the paper results are clearly presented and thoroughly validated, and the takeaways on ViTs helping with (few-shot) OOD identification should be of broad interest to the community.